# A One-Size-Fits-All Approach to Improving Randomness in Paper Assignment

**Yixuan Even Xu**[1]* **Steven Jecmen**[2] **Zimeng Song**[3] **Fei Fang**[2]
[1]Tsinghua University [2]Carnegie Mellon University [3]Independent Researcher
xuyx20@mails.tsinghua.edu.cn
{sjecmen,feif}@cs.cmu.edu
zmsongzm@gmail.com

## Abstract

The assignment of papers to reviewers is a crucial part of the peer review processes of large publication venues, where organizers (e.g., conference program chairs) rely on algorithms to perform automated paper assignment. As such, a major challenge for the organizers of these processes is to specify paper assignment algorithms that find appropriate assignments with respect to various desiderata. Although the main objective when choosing a good paper assignment is to maximize the expertise of each reviewer for their assigned papers, several other considerations make introducing randomization into the paper assignment desirable: robustness to malicious behavior, the ability to evaluate alternative paper assignments, reviewer diversity, and reviewer anonymity. However, it is unclear in what way one should randomize the paper assignment in order to best satisfy all of these considerations simultaneously. In this work, we present a practical, one-size-fits-all method for randomized paper assignment intended to perform well across different motivations for randomness. We show theoretically and experimentally that our method outperforms currently-deployed methods for randomized paper assignment on several intuitive randomness metrics, demonstrating that the randomized assignments produced by our method are general-purpose.

## 1 Introduction

Peer review is the process in which submissions (such as scientific papers) are evaluated by expert reviewers. It is considered a critical part of the scientific process and is commonly used to determine which papers get published in journals and conferences. For concreteness, we set this work in the academic conference setting, although the approach can be generalized to other settings, such as peer review for grant proposals and peer grading in classrooms. Due to the large scale of modern conferences like NeurIPS and AAAI, conference program chairs work closely with assignment algorithms to assign papers to reviewers automatically. Among many other challenges involved in managing huge numbers of reviewers and submissions, these organizers are faced with the difficult task of balancing various considerations for the paper assignment. Human-friendly automated paper assignment algorithms are thus crucial for helping them find desirable paper assignments.

In a standard paper assignment setting, a set $\mathcal{P}$ of $n_p$ papers need to be assigned to a set $\mathcal{R}$ of $n_r$ reviewers. To ensure each paper gets enough reviewers and no reviewer is overloaded with papers, each paper $p$ in $\mathcal{P}$ should be assigned to $\ell_p$ reviewers and each reviewer $r$ in $\mathcal{R}$ should receive no more than $\ell_r$ papers. An assignment is represented as a binary matrix $\mathbf{x}$ in $\{0, 1\}^{n_p \times n_r}$, where $x_{p,r} = 1$ indicates that paper $p$ is assigned to reviewer $r$. The main objective of paper assignment is usually to maximize the predicted match quality between reviewers and papers [1]. To characterize this,

---

*This work was done when Xu was a visiting intern at Carnegie Mellon University.

37th Conference on Neural Information Processing Systems (NeurIPS 2023).

a similarity matrix $\mathbf{S}$ in $\mathbb{R}_{\geq 0}^{n_p \times n_r}$ is commonly assumed [1–7]. Here, $S_{p,r}$ represents the predicted quality of review from reviewer $r$ for paper $p$ and is generally computed from various sources [8]: reviewer and paper subject areas, reviewer-selected bids, and textual similarity between the paper and the reviewer's past work [1, 9–12]. Then, the **quality** of an assignment can be defined as the total similarity of all assigned paper reviewer pairs, i.e., $\mathrm{Quality}(\mathbf{x}) = \sum_{p,r} x_{p,r} S_{p,r}$. One standard approach for computing a paper assignment is to maximize quality [1, 4–7, 13] (which we will refer to as the maximum-quality assignment). Variants of this approach have been widely used by existing conferences, such as NeurIPS, AAAI, and ICML [8].

While the deterministic maximum-quality assignment is the most common, there are strong reasons to introduce randomness into paper assignment – that is, to determine a probability distribution over feasible deterministic assignments and sample one assignment from the distribution. Specifically, randomized paper assignments are beneficial due to the following motivations:

- **Motivation 1: Robustness to malicious behavior.** Several computer science conferences have uncovered "collusion rings" of reviewers and authors [14, 15], in which the reviewers aim to get assigned to the authors' papers in order to give them good reviews without considering their merits. By manipulating their stated expertise and interest (e.g., in the "paper bidding" process), these reviewers can cause the assignment algorithm to believe that their match quality with the targeted papers is very high. In other cases, reviewers may target assignment to a paper with the aim of giving it an unfair negative review [16–18]. Randomization can reduce the probability that a malicious reviewer achieves assignment to a target paper.

- **Motivation 2: Evaluation of alternative assignments.** Accurate reviewer-paper similarity scores are fundamental for automated paper assignment algorithms. Despite this, these scores are currently computed using various different methods by different conferences [19, 20], with no obvious way to tell beforehand which method produces the best-quality reviews. However, after deploying an assignment, it is possible to examine the resulting reviews to counterfactually evaluate the review quality produced by another method of similarity computation that is not deployed. Specifically, using techniques for off-policy evaluation [21], the randomness of the deployed paper assignment can be utilized to estimate the review quality of another non-deployed assignment. The variance of the estimation depends on the overlap in assignment probability between the deployed (randomized) assignment and the non-deployed alternative assignments of interest. Such an evaluation can then inform program chairs on how similarities should be computed (or how other algorithmic choices should be made) in the future.

- **Motivation 3: Reviewer diversity.** As each paper is evaluated by multiple reviewers, it is often desirable to assign a set of reviewers with diverse perspectives or areas of expertise. However, since maximum-quality assignments compute only a holistic score to represent the expertise of each reviewer-paper pair, they do not consider this factor. Randomization can increase diversity by spreading out assignment probability among a larger set of high-expertise reviewers.

- **Motivation 4: Reviewer anonymity.** In peer review, reviewer identities are hidden from authors so that authors cannot retaliate for negative reviews. As a result, conferences are generally reluctant to release paper assignment data since authors may be able to deduce the identities of their reviewers (even if reviewer names and other information are hidden). By sufficiently randomizing the assignment, conferences can make it difficult for authors to identify any reviewer on their paper with high probability from the assignment data.

Despite the significance of randomness in paper assignment, there is very limited prior work that looks into computing randomized assignments. A notable exception is [22], which proposed an algorithm for computing randomized paper assignments: **Probability Limited Randomized Assignment (PLRA)**. Formally, it represents a randomized assignment as a matrix $\mathbf{x}$ in $[0, 1]^{n_p \times n_r}$, where $x_{p,r}$ denotes the marginal probability that paper $p$ is assigned to reviewer $r$. PLRA computes a randomized assignment via the following linear program (LP), defined for a given parameter $Q \in [0, 1]$ as:

$$
\begin{array}{lll}
\text{Maximize} & \mathrm{Quality}(\mathbf{x}) = \sum_{p,r} x_{p,r} S_{p,r} \\
\text{Subject to} & \sum_r x_{p,r} = \ell_p & \forall p \in \mathcal{P}, \\
& \sum_p x_{p,r} \leq \ell_r & \forall r \in \mathcal{R}, \qquad \text{(PLRA)} \\
& 0 \leq x_{p,r} \leq Q & \forall p \in \mathcal{P}, r \in \mathcal{R},
\end{array}
$$

where $\mathrm{Quality}(\mathbf{x})$ for a randomized assignment $\mathbf{x}$ is the expected total similarity of the assignment. A deterministic assignment can then be sampled, using the fact that any feasible randomized assignment $\mathbf{x}$ can be implemented as a distribution over feasible deterministic assignments [22, 23].

PLRA is primarily concerned with the first motivation for randomization: robustness to malicious behavior. By limiting each entry of $\mathbf{x}$ to be at most $Q$, PLRA guarantees that any malicious reviewer aiming to be assigned to a target paper has at most probability $Q$ to succeed, even if the reviewer and paper are chosen adversarially. The hyperparameter $Q$ can be adjusted to balance the loss in quality and level of randomization. PLRA has been deployed in multiple iterations of the AAAI conference [8] and is implemented at the popular conference management system OpenReview.net [24].

However, PLRA does not fully solve the problem of randomized paper assignment. In particular, PLRA is specific to one metric of randomness: the maximum assignment probability across all paper-reviewer pairs. As a result, it is not clear how well PLRA aligns with motivations for randomization other than robustness to malicious behavior. Moreover, PLRA does not distinguish between different solutions with the same quality and maximum assignment probability. This means that it often loses opportunities to add additional randomness since it neglects to consider non-maximum assignment probabilities. One way to remedy this issue is to allow different values of $Q$ to be set for each pair $(p, r)$, as in the original formulation of [22]. However, this level of flexibility makes it a significant burden for program chairs to manually choose appropriate $Q$ values, hindering the usability of the algorithm. As a result, only the single-$Q$ version stated above has been deployed in practice.

In this work, we address the problem of randomizing paper assignment by looking for a simple and practical method of achieving general-purpose randomized paper assignments. We consider various intuitive randomness metrics that are relevant to all above motivations but not overly specific to a particular problem formulation, and aim to provide a method that performs well across these metrics. In this way, we can provide a method for randomized paper assignment that conference program chairs can easily deploy without needing to precisely specify objectives or hyperparameters.

More specifically, we make the following contributions in this work. **(1).** We define several metrics to measure the extent to which the randomization in a randomized paper assignment satisfies the stated motivations (Section 3). **(2).** We propose Perturbed Maximization (PM), a practical algorithm for randomized paper assignment that does not rely on any specific formulation of the stated motivations (Section 4). While our algorithm can be implemented using a standard convex optimization solver, we additionally propose an approximate implementation that is computationally cheaper. **(3).** We provide theoretical results showing that PM provably outperforms PLRA on two classes of structured similarity scores (Section 5). **(4).** We extensively evaluate our algorithm via experiments using two realistic datasets of similarity scores from AAMAS 2015 and ICLR 2018 (Section 6). PM simultaneously performs well on all defined randomness metrics while sacrificing a small amount of quality as compared to the optimal non-randomized assignment. Additionally, our experiments show that PM achieves good performance when hyperparameters are set based only on the desired assignment quality, ensuring that it is simple for program chairs to deploy in practice.

## 2 Related Work

This work follows a recent area of research in computer science on paper assignment algorithms for peer review. Building on the standard approach of maximum-similarity assignment, algorithms have been proposed to handle various additional considerations in the paper assignment: the fairness across papers [2], the seniority of reviewers [20], review processes with multiple phases [3], strategyproofness [25, 26], and many others [8]. We note here specifically two relevant lines of work.

One motivation for our randomized assignment algorithm is to provide robustness to malicious reviewers, a problem considered by several past works [27, 28]. Although our work most closely relates to the randomized paper assignment algorithm proposed in [22], other non-randomized approaches to the problem exist. Leyton-Brown et al. [20] describe the paper assignment process used at AAAI 2021, which included several additional soft constraints intended to curb the possibility of reviewer-author collusion rings. Wu et al. [29] propose fitting a model of reviewer bidding in order to smooth out irregular bids. Boehmer et al. [30] consider computing paper assignments without short-length cycles in order to prevent quid-pro-quo agreements between reviewers.

Randomization in the paper assignment process has also been used for evaluating different assignment policies. Traditionally, conferences will sometimes run randomized controlled experiments in order to test policy changes, where the randomization is incorporated in the assignment of reviewers to different experimental conditions. For example, WSDM 2017 randomly separated reviewers into single- and double-blind conditions in order to evaluate the benefits of double-blind reviewing [31]; other notable experiments include NeurIPS 2014 [32, 33], ICML 2020 [34], and NeurIPS 2021 [35]. In contrast, recent work by Saveski et al. [21] proposes a method for evaluating alternative assignment policies by leveraging randomization in the paper assignment itself, such as the randomized paper assignments of [22]. By introducing additional randomness at a low cost, our algorithm provides an improved basis for the methods of [21] and potential future counterfactual policy evaluation methods.

## 3 Metrics for Randomness and Problem Statement

In Section 1, we introduced several motivations for choosing randomized paper assignments: robustness to malicious behavior, evaluation of alternative assignments, reviewer diversity, and reviewer anonymity. These indicate that assignment quality is not the only objective that should be considered when choosing a paper assignment. In this section, we propose several metrics to characterize the extent to which the randomness in a paper assignment is practically useful.

One randomness metric considered by PLRA is **maximum probability**, defined as $\mathrm{Maxprob}(\mathbf{x}) = \max_{p,r}\{x_{p,r}\}$. PLRA controls $\mathrm{Maxprob}$ in order to trade off between quality and randomness, and thus achieves the greatest possible quality for a fixed level of $\mathrm{Maxprob}$. However, as the following example shows, this metric alone is not sufficient to fully characterize the randomness of a paper assignment since it ignores the structure of the assignment with non-maximum probability.

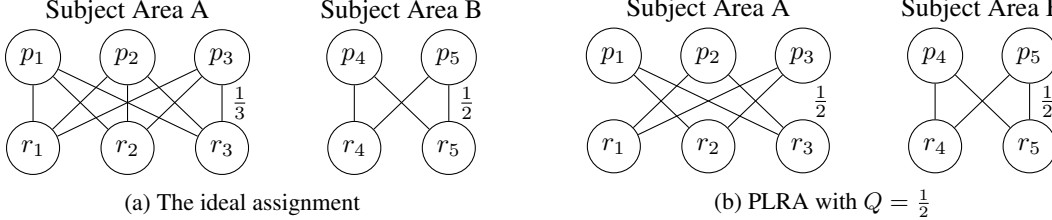

(a) The ideal assignment          (b) PLRA with $Q = \frac{1}{2}$

Figure 1: Example showing the limitations of PLRA. There are 2 subject areas with 5 papers and 5 reviewers. $\ell_p = \ell_r = 1$. The similarity of paper-reviewer pair is 1 if they are in the same area. The edge weights denote the assignment probability. (a) shows the ideal assignment with optimal quality. (b) shows an assignment by PLRA with $Q = \frac{1}{2}$, which is less randomized than (a) in subject area A.

Fig. 1 depicts a mini-conference consisting of 2 subject areas. Subject area A contains 3 papers and 3 reviewers, whereas subject area B contains 2 of each. The similarities between reviewers and papers within the same subject area are 1 and similarities across subject areas are 0. The constraints are $\ell_p = \ell_r = 1$, i.e., one-to-one assignment. Intuitively, the best randomized assignment matches papers and reviewers uniformly at random within each subject area, like Fig. 1a, since this does not sacrifice any quality. However, PLRA fails to give out this ideal assignment regardless of the hyperparameter choice. Specifically, $Q \geq 1/2$ is required for PLRA to get a maximum-quality assignment, but when $Q \geq 1/2$, PLRA considers Fig. 1a and Fig. 1b to be equivalent (in terms of objective value). In fact, infinitely many solutions with the same optimal quality are considered equivalent by PLRA, and current LP solvers tend to return Fig. 1b as it is a vertex solution when $Q = 1/2$. In essence, PLRA is leaving "free randomness on the table," with many practical implications.

Consider the problem of mitigating malicious behavior (Motivation 1), and compare the assignments in Figs. 1a and 1b. Although they appear to have the same $\mathrm{Maxprob}$, when we only look at subject area A, the $\mathrm{Maxprob}$ of Fig. 1a is lower than that of Fig. 1b. This indicates that Fig. 1a is more robust to malicious behavior within subject area A. Therefore, Fig. 1a is more desirable.

Moreover, as introduced in Motivation 2, randomness can also be leveraged to evaluate alternative paper assignments using observations of the review quality from a deployed, randomized assignment. These techniques rely on a "positivity" assumption: paper-reviewer pairs assigned in the alternative assignment must be given non-zero probability in the deployed assignment. Thus, if we spread assignment probability more uniformly among more reviewer-paper pairs, the resulting data can be

used to evaluate more varied strategies with tighter bounds. In this sense, the assignment in Fig. 1a will allow us to better estimate the quality of different paper assignments within subject area A.

The failure of PLRA on such a simple example shows that $\mathrm{Maxprob}$ alone is an inadequate metric. Thus, we need other metrics to distinguish between assignments like Fig. 1a and Fig. 1b. We therefore propose, in addition to $\mathrm{Maxprob}$, a set of new randomness metrics to capture the neglected low-probability structure of an assignment. Under each of these metrics, a uniform assignment is considered "more random" than any other assignment, thus distinguishing Fig. 1a from Fig. 1b.

(1) **Average maximum probability:** $\mathrm{AvgMaxp}(\mathbf{x}) = \frac{1}{n_p} \sum_p \max_r \{x_{p,r}\}$. With respect to the motivation of preventing malicious behavior, this randomness metric corresponds to the case when a target paper is randomly chosen and the reviewer targeting assignment to that paper is adversarially chosen. By minimizing average maximum probability, we will limit the success probability of manipulation in that case.

(2) **Support size:** $\mathrm{Support}(\mathbf{x}) = \sum_{p,r} \mathbb{I}[x_{p,r} > 0]$. Support size directly relates to the "positivity" assumption introduced above. As there may be many alternative paper assignments of interest, maximizing the support size effectively maximizes the quality of estimation across them.

(3) **Entropy:** $\mathrm{Entropy}(\mathbf{x}) = -\sum_{p,r} x_{p,r} \ln(x_{p,r})$. In information theory, entropy characterizes the uncertainty of a random variable. By maximizing entropy, we maximize the uncertainty of our assignment, corresponding to the idea of maximizing randomness. Note that strictly speaking, assignment $\mathbf{x}$ is not a probability distribution, so this definition is a generalization.

(4) **L2 norm:** $\mathrm{L2Norm}(\mathbf{x}) = \sqrt{\sum_{p,r} x_{p,r}^2}$. To prevent manipulation, PLRA limits the assigned probability of each pair to be at most a specified value $Q$. L2 norm relaxes this constraint to a soft one: a higher probability results in a larger loss. Note that a uniformly random assignment will always have the smallest L2 norm, and so minimizing L2 norm pushes the assignment towards the uniform assignment.

The combination of these metrics more comprehensively captures the impact of randomness on the motivations from Section 1. Moreover, they do so without requiring a specific problem formulation for each motivation (e.g., an assumption on the behavior of malicious reviewers, a list of the alternative assignments of interest), which are impractical or infeasible to accurately specify in practice.

**Problem statement.** For an input instance $(n_p, n_r, \ell_p, \ell_r, \mathbf{S})$, we want to find an algorithm achieving a good trade-off between quality and randomness. Specifically, let the maximum possible quality be $M$. For a given lower bound of assignment quality $\eta M$ ($\eta \in [0, 1]$), we want the algorithm to produce an assignment $\mathbf{x}$ with lower $\mathrm{Maxprob}(\mathbf{x}), \mathrm{AvgMaxp}(\mathbf{x}), \mathrm{L2Norm}(\mathbf{x})$ and higher $\mathrm{Entropy}(\mathbf{x}), \mathrm{Support}(\mathbf{x})$, i.e., a good Pareto-frontier of quality and randomness.

## 4   Perturbed Maximization

In this section, we present our proposed algorithm for randomized paper assignment. To describe it, we first present some definitions.

**Definition 4.1** (Perturbation Function). *A function $f : [0, 1] \to [0, 1]$ is a **perturbation function** if (i) $f(0) = 0$, (ii) $f'$ exists, (iii) $f$ is non-decreasing on $[0, 1]$ and (iv) $f$ is concave on $[0, 1]$.*

**Definition 4.2** (Perturbed Quality). *For an assignment $\mathbf{x}$, its **perturbed quality** with respect to perturbation function $f$ on instance $(n_p, n_r, \ell_p, \ell_r, \mathbf{S})$ is $\mathrm{PQuality}_f(\mathbf{x}) = \sum_{p,r} S_{p,r} \cdot f(x_{p,r})$.*

The definition of perturbed quality incorporates the intuition from the motivating example in the previous section. As function $f$ is concave, the marginal increase of $\mathrm{PQuality}_f(\mathbf{x})$ from increasing a specific entry $x_{p,r}$ is diminishing as $x_{p,r}$ grows. Consequently, the assignment of Fig. 1a will have a higher perturbed quality than that of Fig. 1b. This naturally gives our new algorithm, **Perturbed Maximization (PM)**. For a given parameter $Q \in [0, 1]$ and a perturbation function $f$:

$$
\begin{array}{lll}
\text{Maximize} & \mathrm{PQuality}_f(\mathbf{x}) = \sum_{p,r} S_{p,r} \cdot f(x_{p,r}) & \\
\text{Subject to} & \sum_r x_{p,r} = \ell_p & \forall p \in \mathcal{P}, \\
& \sum_p x_{p,r} \leq \ell_r & \forall r \in \mathcal{R}, \quad \text{(PM)} \\
& 0 \leq x_{p,r} \leq Q & \forall p \in \mathcal{P}, r \in \mathcal{R}.
\end{array}
$$

Note that PM is a class of algorithms induced by different perturbation functions. When the perturbation function $f$ is chosen to be a linear function, PM becomes PLRA. In the main experiments of this paper, two specific perturbation functions are considered: **(1). Exponential Perturbation Function:** $f(x) = 1 - e^{-\alpha x}$ where $\alpha \in (0, +\infty)$ and **(2). Quadratic Perturbation Function:** $f(x) = x - \beta x^2$ where $\beta \in [0, 1]$. Respectively, we will denote PM with $f(x) = 1 - e^{-\alpha x}$ and $f(x) = x - \beta x^2$ as **PM-Exponential (PM-E)** and **PM-Quadratic (PM-Q)** in the rest of the paper.

In Section 6, we will empirically show that under our settings of hyperparameters, PM-E and PM-Q have almost identical performances on every metric we consider, which suggests that the specific form of the perturbation function has limited impact as long as it is strictly concave. Therefore, there is likely no need to consider many different types of perturbation functions.

To analyze PM, first notice that the concaveness of function $f$ guarantees that the optimization program is concave. Therefore, we can use standard concave optimization methods like gradient ascent or the ellipsoid method to solve the program in polynomial time. In most of the experiments of this paper, we will use Gurobi [36], a well-known commercial solver, to solve PM. While solving PM as a general concave optimization problem is conceptually convenient, doing so also incurs a high time complexity as there are $n_p \cdot n_r$ variables. To further speed up the execution of PM, we propose Algorithm 1, a network-flow-based approximation of PM.

---

**Algorithm 1**: Network-Flow-Based Approximation of PM

For a given parameter $Q \in [0, 1]$, a perturbation function $f$ and a precision $w \in \mathbb{N}^+$:

(1) Construct a graph $G$ with a source $s$, a sink $t$.

(2) For paper $p$, add a vertex $v_p$ and an edge $s \to v_p$ with capacity $\ell_p \cdot w$ and cost 0.

(3) For reviewer $r$, add a vertex $v_r$ and an edge $v_r \to t$ with capacity $\ell_r \cdot w$ and cost 0.

(4) For a reviewer-paper pair $(p, r)$, add $\lfloor Q \cdot w \rfloor$ edges $v_p \to v_r$. The $i$-th edge has capacity 1 and cost $S_{p,r} \cdot [f(\frac{i}{w}) - f(\frac{i-1}{w})]$. Let this set of edges be $E_{p,r}$.

(5) Run maximum cost maximum flow algorithm [37] on $G$. Define the assignment $\mathbf{x}$ such that $x_{p,r} = [\text{The total flow on edges in } E_{p,r}]/w$.

(6) Enumerate all pairs $(p, r)$ in any order and increase $x_{p,r}$ by $\min\{Q - x_{p,r}, \ell_p - \sum_r x_{p,r}, \ell_r - \sum_p x_{p,r}\}$. If $\sum_{p,r} x_{p,r} = n_p \cdot \ell_p$, return $\mathbf{x}$. Otherwise, return infeasible.

---

At a high level, Algorithm 1 uses a piecewise linear function with $w$ pieces to approximate the concave function $f(x)$ and solves the approximated objective with maximum cost maximum flow. As we increase the precision $w$, the approximation becomes more accurate, but the running time of the algorithm also scales up. Formally, we have the following Theorem 1.

**Theorem 1.** *Let $w \in \mathbb{N}^+$ be the precision in Algorithm 1, and* OPT *be the optimal perturbed quality.*

(a) *Algorithm 1 runs in $O(w \cdot \ell_p \cdot n_p{}^2 \cdot n_r)$ time.*

(b) *Algorithm 1 returns an assignment with perturbed quality* ALG $\geq$ OPT $- f(\frac{1}{w}) \sum_{p,r} S_{p,r}$.

The proof of Theorem 1 is deferred to Appendix B.1.

Consider the running time of Algorithm 1 given in Theorem 1 (a). If we directly model PM as an optimization problem, the number of variables will be $n = n_p \cdot n_r$. The state-of-the-art algorithm for solving a linear program with $n$ variables to high accuracy has a time complexity of $O^*(n^{2.38} \log n)$ [38]. In contrast, for fixed $w$, Algorithm 1 works in $O(n^2)$ time since $\ell_p \leq n_r$. Thus, Algorithm 1 has a better time complexity than directly solving PM as an optimization program even if the objective is linear. For non-linear perturbation functions, the time complexity of Algorithm 1 remains the same while the complexity of solving the optimization program increases. Moreover, by Theorem 1 (b), we can see that as $w$ increases, the approximated perturbed quality ALG approaches OPT, formalizing the intuition that as we increase the precision $w$, the approximation becomes more accurate. In Section 6, we empirically evaluate the running time and the approximation accuracy of Algorithm 1. We find with $w = 10$, Algorithm 1 produces decently-accurate approximations on our datasets.

# 5 Theoretical Analysis

In this section, we provide two theorems showing that PM provably outperforms PLRA on a general class of input instances. We start with the simpler one inspired by the example in Fig. 1. Let $\mathrm{PLRA}(Q)$ and $\mathrm{PM}(Q, f)$ be the set of possible solutions of PLRA and PM respectively.

**Definition 5.1** (Blockwise Dominant Matrix). *A similarity matrix* $\mathbf{S} \in \mathbb{R}_{\geq 0}^{n_p \times n_r}$ *is **blockwise dominant** with **block identity** $\mathbf{A} \in \mathbb{R}_{\geq 0}^{k \times k}$ and **block sizes** $\{p_1, \ldots, p_k\}, \{r_1, \ldots, r_k\}$ if*

$$\mathbf{S} = \begin{pmatrix} A_{1,1} \cdot \mathbf{1}_{p_1 \times r_1} & A_{1,2} \cdot \mathbf{1}_{p_1 \times r_2} & \ldots & A_{1,k} \cdot \mathbf{1}_{p_1 \times r_k} \\ A_{2,1} \cdot \mathbf{1}_{p_2 \times r_1} & A_{2,2} \cdot \mathbf{1}_{p_2 \times r_2} & \ldots & A_{2,k} \cdot \mathbf{1}_{p_2 \times r_k} \\ \ldots & \ldots & \ldots & \ldots \\ A_{k,1} \cdot \mathbf{1}_{p_k \times r_1} & A_{k,2} \cdot \mathbf{1}_{p_k \times r_2} & \ldots & A_{k,k} \cdot \mathbf{1}_{p_k \times r_k} \end{pmatrix},$$

*where $A_{i,i} > A_{i,j}, \forall i \neq j$ and $\mathbf{1}_{p \times r}$ is a $p \times r$ matrix with all entries being 1. Moreover, define the **dominance factor** of $\mathbf{A}$ as $\mathrm{Dom}(\mathbf{A}) = \sup\{\alpha \mid A_{i,i} \geq \alpha \cdot A_{i,j}, \forall i \neq j\}$.*

**Remark.** Like Fig. 1, a blockwise dominant similarity matrix models a conference with $k$ subject areas where the $i$-th subject area has $p_i$ papers and $r_i$ reviewers. The similarity between a paper and a reviewer is determined only by their subject areas, and a paper has highest similarity with a reviewer in the same subject area. Note that $\mathrm{Dom}(\mathbf{A}) > 1$ as $A_{i,i} > A_{i,j}, \forall i \neq j$.

**Theorem 2.** *For an input instance $(n_p, n_r, \ell_p, \ell_r, \mathbf{S})$, where $\mathbf{S}$ is blockwise dominant with block identity $\mathbf{A} \in \mathbb{R}_{\geq 0}^{k \times k}$ and block sizes $\{p_1, \ldots, p_k\}, \{r_1, \ldots, r_k\}$, assume (i) $\{r_1, \ldots, r_k\}$ are not all equal, (ii) $p_i \cdot \ell_p \leq r_i \cdot \ell_r \; \forall i \in \{1, \ldots, k\}$ and (iii) $Q \cdot r_i \geq \ell_p \; \forall i \in \{1, \ldots, k\}$. Let $f$ be a strictly concave perturbation function and $f'(0) < \mathrm{Dom}(\mathbf{A}) f'(1)$. PM with $f(x)$ as the perturbation function (weakly) dominates PLRA in quality and all randomness metrics. Formally,*

(a) $\forall \mathbf{x} \in \mathrm{PM}(Q, f), \forall \mathbf{y} \in \mathrm{PLRA}(Q),$

$$\mathrm{Quality}(\mathbf{x}) \geq \mathrm{Quality}(\mathbf{y}), \quad \mathrm{Maxprob}(\mathbf{x}) \leq \mathrm{Maxprob}(\mathbf{y}), \quad \mathrm{AvgMaxp}(\mathbf{x}) \leq \mathrm{AvgMaxp}(\mathbf{y}),$$
$$\mathrm{L2Norm}(\mathbf{x}) \leq \mathrm{L2Norm}(\mathbf{y}), \quad \mathrm{Entropy}(\mathbf{x}) \geq \mathrm{Entropy}(\mathbf{y}), \quad \mathrm{Support}(\mathbf{x}) \geq \mathrm{Support}(\mathbf{y}).$$

(b) $\forall \mathbf{x} \in \mathrm{PM}(Q, f), \exists \mathbf{y} \in \mathrm{PLRA}(Q),$

$$\mathrm{AvgMaxp}(\mathbf{x}) < \mathrm{AvgMaxp}(\mathbf{y}), \quad \mathrm{L2Norm}(\mathbf{x}) < \mathrm{L2Norm}(\mathbf{y}), \quad \mathrm{Entropy}(\mathbf{x}) > \mathrm{Entropy}(\mathbf{y}).$$

The proof of Theorem 2 follows the same intuition as the example in Fig. 1. The assumptions guarantee that an assignment like Fig. 1a, i.e., uniformly matching papers to reviewers within the same subject area, is feasible. Details of the proof are deferred to Appendix B.2. At a high level, Theorem 2 shows that with a slight restriction on the perturbation function, PM provably performs better than PLRA on blockwise dominant similarity matrices.

Theorem 2 requires a strict restriction on $\mathbf{S}$'s structure. In our next theorem, we remove the restriction, stating that PM also outperforms PLRA on a random similarity matrix with high probability.

**Theorem 3.** *For an input instance $(n_p, n_r, \ell_p, \ell_r, \mathbf{S})$, where each entry of $\mathbf{S}$ is i.i.d. sampled from $\{v_1, \ldots, v_k\}$ $(0 < v_1 < \cdots < v_k)$ uniformly, assume (i) $k \leq \frac{1}{c} \cdot n_p$, (ii) $\ell_r \geq c \cdot \ln(n_r)$, (iii) $2 \cdot n_p \cdot \ell_p \leq n_r \cdot \ell_r$ and (iv) $Q \cdot (n_r - 1) \geq \ell_p$. Let $f$ be a strictly concave perturbation function and $f'(0) < \frac{v_i}{v_{i-1}} f'(1), \forall i \in \{2, \ldots, k\}$. With probability $1 - e^{-\Omega(c)}$, PM with $f(x)$ as the perturbation function (weakly) dominates PLRA in quality and all randomness metrics. Formally,*

(a) $\forall \mathbf{x} \in \mathrm{PM}(Q, f), \forall \mathbf{y} \in \mathrm{PLRA}(Q),$

$$\mathrm{Quality}(\mathbf{x}) \geq \mathrm{Quality}(\mathbf{y}), \quad \mathrm{Maxprob}(\mathbf{x}) \leq \mathrm{Maxprob}(\mathbf{y}), \quad \mathrm{AvgMaxp}(\mathbf{x}) \leq \mathrm{AvgMaxp}(\mathbf{y}),$$
$$\mathrm{L2Norm}(\mathbf{x}) \leq \mathrm{L2Norm}(\mathbf{y}), \quad \mathrm{Entropy}(\mathbf{x}) \geq \mathrm{Entropy}(\mathbf{y}), \quad \mathrm{Support}(\mathbf{x}) \geq \mathrm{Support}(\mathbf{y}).$$

(b) $\forall \mathbf{x} \in \mathrm{PM}(Q, f), \exists \mathbf{y} \in \mathrm{PLRA}(Q),$

$$\mathrm{Support}(\mathbf{x}) > \mathrm{Support}(\mathbf{y}), \quad \mathrm{L2Norm}(\mathbf{x}) < \mathrm{L2Norm}(\mathbf{y}), \quad \mathrm{Entropy}(\mathbf{x}) > \mathrm{Entropy}(\mathbf{y}).$$

The complete proof of Theorem 3 is deferred to Appendix B.3. To sketch the proof, we will first relate PLRA and PM to two simpler auxiliary algorithms using a concentration inequality and then

prove the dominance between them. The assumptions (ii) and (iii) are used to connect PM and PLRA with the auxiliary algorithms, while (i) and (iv) are for proving the dominance.

For Theorem 3 to hold on general random similarity matrices, we have made a major assumption (i), which requires that there are not too many distinct levels of similarity scores. Assumption (i) is naturally satisfied in some cases. For example, when the similarities are derived purely from reviewers' bids, the number of levels becomes the number of discrete bid levels; discrete values can also result when computing similarities from the overlap between reviewer- and author-selected subject areas. When the similarity scores are instead computed from continuous values like TPMS scores [1], assumption (i) is usually not satisfied. In Section 6, we show experiments on both discrete and continuous similarities to evaluate the effect of assumption (i).

## 6 Experiments

**Datasets.** In this section, we test our algorithm on two realistic datasets. The first dataset is bidding data from the AAMAS 2015 conference [39]. In this dataset, $n_p = 613, n_r = 201$, and the bidding data has 4 discrete levels: "yes", "maybe", "no" and "conflict". We transform these 4 levels to similarities $1, \frac{1}{2}, \frac{1}{4}$ and 0, as inspired by NeurIPS 2016, in which similarity scores were computed as $2^{\text{bid}}(0.5s_{\text{text}} + 0.5s_{\text{subject}})$ [19]. The transformed dataset satisfies assumption (i) in Theorem 3 as there are only 4 levels. The second dataset contains text-similarity scores recreated from the ICLR 2018 conference with $n_p = 911, n_r = 2435$ [26]. These scores were computed by comparing the text of each paper with the text of each reviewer's past work; we directly use them as the similarity matrix. Assumption (i) is not valid in this dataset as it is continuous-valued. The constraints are set as $\ell_p = 3, \ell_r = 6$ for ICLR 2018 as was done in [26] and $\ell_p = 3, \ell_r = 12$ for AAMAS 2015 for feasibility. In Appendix A.3, we also test our algorithm on four additional datasets from [39].

**Experiment and hyperparameter setting.** We implement PLRA and two versions of PM (PM-E and PM-Q) using commercial optimization solver Gurobi 10.0 [36]. For each algorithm on each dataset, we use a principled method (Appendix A.2) to find 8 sets of hyperparameters that produce solutions with at least $\{80\%, 85\%, 90\%, 95\%, 98\%, 99\%, 99.5\%, 100\%\}$ of the maximum possible quality. When setting the hyperparameters, we choose a "slackness" value $\delta$ and allow PM to produce solutions with $\text{Maxprob}$ at most $\delta$ higher than the optimal $\text{Maxprob}$ (in exchange for better performance on other metrics). For AAMAS 2015, $\delta = 0.00$ and for ICLR 2018, $\delta = 0.02$. We refer to Appendix A.2 for more details. We do not consider assignments with lower than $80\%$ of the maximum quality since such low-quality assignments are unlikely to be deployed in practice. We evaluate the produced assignments on different randomness metrics to draw Figs. 2 to 4. We also implemented Algorithm 1 and tested it with the set of parameters at $95\%$ relative quality to create Table 1. All source code is released at `https://github.com/YixuanEvenXu/perturbed-maximization`, and all experiments are done on a server with 56 cores and 504G RAM, running Ubuntu 20.04.6.

| Implementation | Gurobi | | Flow ($w = 10$) | |
|---|---|---|---|---|
| Algorithm | PM-Q | PM-E | PM-Q | PM-E |
| Wall Clock Time | 86.85s | 368.13s | 563.24s | 554.07s |
| Total CPU Time | 988.11s | 3212.18s | 562.77s | 553.59s |
| Quality | 95% | 95% | 94% | 94% |
| PQuality | 606.55 | 305.41 | 603.34 | 303.68 |
| AvgMaxp ($\downarrow$) | 0.87 | 0.87 | 0.86 | 0.86 |
| Entropy ($\uparrow$) | 1589.95 | 1630.80 | 1575.86 | 1607.20 |

Table 1: Comparison of different implementations on ICLR 2018. Downarrows ($\downarrow$) mean the lower the better and uparrows ($\uparrow$) mean the higher the better. Due to parallelization, Gurobi takes less wall clock time. However, the network-flow-based approximation uses fewer computation resources (total CPU time). The approximation has relatively accurate $\text{Quality}$, $\text{PQuality}$ and randomness metrics.

**Comparing Gurobi and Algorithm 1.** Table 1 shows that on ICLR 2018, Gurobi takes less wall clock time than Algorithm 1 due to parallelization over the server's 112 cores. However, Algorithm 1 takes less total CPU time (system + user CPU time) and achieves similar performance with Gurobi, which shows that Algorithm 1 provides decently approximated solutions using fewer computation

resources. Due to the space limit, rows in Table 1 have been selected and only results on ICLR 2018 are present. The complete tables and analysis for both datasets can be found in Appendix A.1.

**Comparing randomness metrics of PM and PLRA on discrete-valued datasets.** As shown in Figs. 2a and 3, on AAMAS 2015, both versions of PM achieve exactly the same performance with PLRA on $\mathrm{Maxprob}$ while improving significantly on the other randomness metrics. Recall that PLRA achieves the optimal quality for a given $\mathrm{Maxprob}$. This demonstrates that as predicted by Theorem 3, PM produces solutions that are more generally random while preserving the optimality in $\mathrm{Maxprob}$ on discrete-valued datasets where assumption (i) holds. In Appendix A.3, we also test our algorithm on four more datasets from [39] and observe similar results.

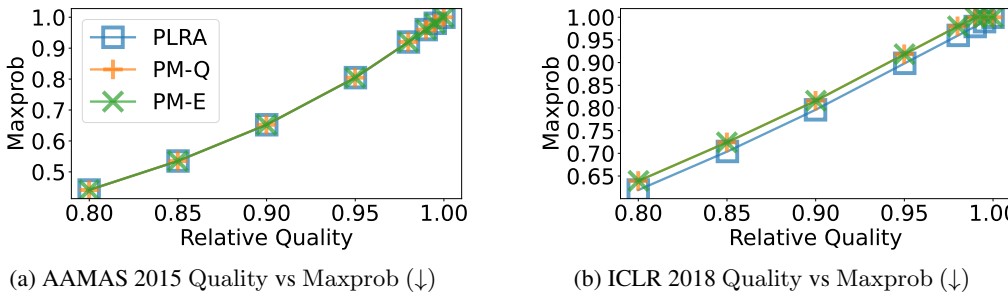

(a) AAMAS 2015 Quality vs Maxprob ($\downarrow$)    (b) ICLR 2018 Quality vs Maxprob ($\downarrow$)

Figure 2: The trade-offs between quality and $\mathrm{Maxprob}$ on both datasets. Downarrows ($\downarrow$) indicate that lower is better. On AAMAS 2015 where assumption (i) of Theorem 3 holds, PM incurs no increase in $\mathrm{Maxprob}$; on ICLR 2018 where it does not hold, PM causes a $0.02$ increase.

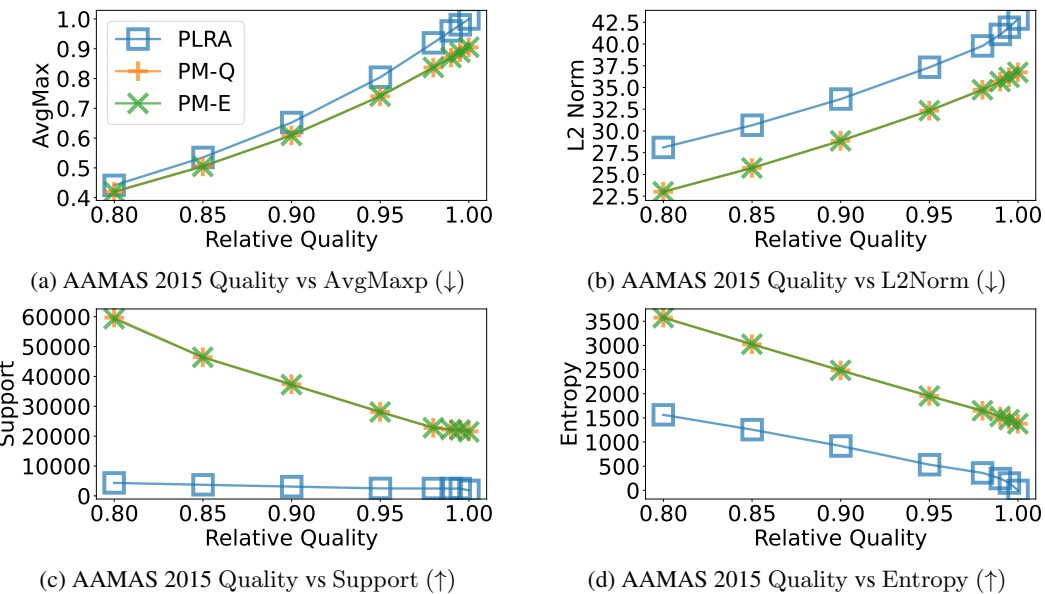

(a) AAMAS 2015 Quality vs AvgMaxp ($\downarrow$)    (b) AAMAS 2015 Quality vs L2Norm ($\downarrow$)

(c) AAMAS 2015 Quality vs Support ($\uparrow$)    (d) AAMAS 2015 Quality vs Entropy ($\uparrow$)

Figure 3: The trade-offs between quality and four randomness metrics on AAMAS 2015. Downarrows ($\downarrow$) indicate that lower is better and uparrows ($\uparrow$) indicate that higher is better. The figure shows that PM significantly outperforms PLRA on all four metrics, as predicted by Theorem 3.

**Comparing randomness metrics of PM and PLRA on continuous-valued datasets.** The results on ICLR 2018 in Figs. 2b and 4 show that both versions of PM sacrifice some $\mathrm{Maxprob}$ as compared to PLRA. The exact amount of this sacrifice is affected by the slackness $\delta$ described in Appendix A.2. However, the improved randomness is still significant, though to a lesser extent than on AAMAS 2015. Note that assumption (i) of Theorem 3 does not hold on ICLR 2018. This exhibits the empirical efficacy of PM beyond the theoretical guarantees provided in Section 5.

**Comparing randomness metrics of PM-E and PM-Q.** In all of Figs. 2 to 4, we can see that no matter what metric and dataset are used, the performances of PM-E and PM-Q are always similar.

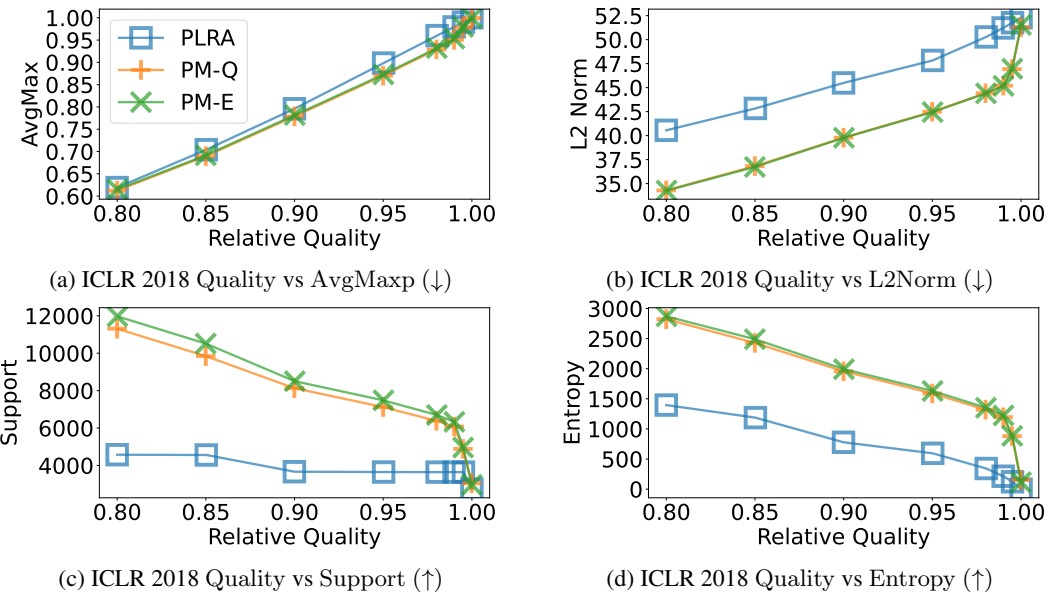

(a) ICLR 2018 Quality vs AvgMaxp ($\downarrow$)

(b) ICLR 2018 Quality vs L2Norm ($\downarrow$)

(c) ICLR 2018 Quality vs Support ($\uparrow$)

(d) ICLR 2018 Quality vs Entropy ($\uparrow$)

Figure 4: The trade-offs between quality and four randomness metrics on ICLR 2018. Downarrows ($\downarrow$) indicate that lower is better and uparrows ($\uparrow$) indicate that higher is better. The figure shows that PM outperforms PLRA on all four metrics despite the fact that assumption (i) of Theorem 3 does not hold, indicating the empirical efficacy of PM beyond theoretical guarantees.

This suggests that the specific type of the perturbation function has limited impact as long as it is strictly concave. Therefore, program chairs do not need to carefully consider this choice in practice.

**Additional experiments.** Due to the page limit on the main paper, we are not able to present all the experiments we run in this section. We refer to Appendix A for additional experimental results about the network-flow-based approximation (Appendix A.1), hyperparameter tuning (Appendix A.2) and experiments on more datasets (Appendix A.3).

## 7   Conclusion

We present a general-purpose, practical algorithm for use by conference program chairs in computing randomized paper assignments. We show both theoretically and experimentally that our algorithm significantly improves over the previously deployed randomized assignment algorithm, PLRA. In conferences that currently deploy PLRA, our algorithm can be simply plugged-in in place of it to find a paper assignment with the same quality but with an improved level of randomization, achieving various benefits.

In practice, various aspects of paper assignment other than quality and randomness are considered by program chairs. We refer to Appendix C for discussions about how to optimize a specific metric with our algorithm (Appendix C.1) and how to incorporate some additional constraints from [20] (Appendix C.2). That being said, we do not consider various other aspects and constraints that may be desired by program chairs: e.g., fairness-based objectives [2] or constraints on review cycles [20]. Incorporating these aspects remains an interesting direction for future work.

**Broader Impacts.** We expect our algorithm to have a mostly positive social impact by assisting program chairs in mitigating malicious behavior, facilitating evaluation of assignments, and increasing reviewer diversity and reviewer anonymity. While deploying a randomized assignment may negatively impact the assignment quality as compared to a deterministic assignment, our work allows conferences to choose the level of randomization that they deem best for their goals.

## Acknowledgments and Disclosure of Funding

This work was supported by ONR grant N000142212181 and NSF grant IIS-2200410.

This work was supported in part by the Sloan Research Fellowship.

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

# A    Additional Experiments

## A.1    Network-Flow-Based Approximation

In Section 6, we presented Table 1, which is a shortened version of the experimental results comparing Gurobi and Algorithm 1. We now present the complete experimental results on both datasets in Tables 2 and 3 as well as the complete analysis in this section.

| Implementation | Gurobi | | | Flow ($w = 10$) | |
|---|---|---|---|---|---|
| Algorithm | PLRA | PM-Q | PM-E | PM-Q | PM-E |
| Wall Clock Time | 1.35s | 4.06s | 17.64s | 20.03s | 20.63s |
| Total CPU Time | 9.91s | 31.13s | 79.20s | 20.03s | 20.61s |
| User CPU Time | 4.94s | 23.46s | 66.44s | 20.03s | 20.61s |
| System CPU Time | 4.97s | 7.67s | 12.76s | 0.00s | 0.00s |
| Quality | 95% | 95% | 95% | 95% | 95% |
| PQuality | N/A | 1232.91 | 312.78 | 1230.75 | 312.22 |
| Maxprob ($\downarrow$) | 0.80 | 0.80 | 0.80 | 0.80 | 0.80 |
| AvgMaxp ($\downarrow$) | 0.80 | 0.74 | 0.74 | 0.74 | 0.74 |
| Support ($\uparrow$) | 2501 | 28108 | 28099 | 5849 | 5853 |
| Entropy ($\uparrow$) | 531.40 | 1953.55 | 1953.20 | 1411.82 | 1411.12 |
| L2Norm ($\downarrow$) | 37.33 | 32.33 | 32.34 | 32.66 | 32.68 |

Table 2: Comparison of different implementations on AAMAS 2015. Downarrows ($\downarrow$) mean the lower the better and uparrows ($\uparrow$) mean the higher the better. Due to parallelization, Gurobi takes less wall clock time. However, the network-flow-based approximation uses fewer computation resources (total CPU time). The approximation has very accurate Quality and PQuality. Approximated Support and Entropy are significantly worse than Gurobi's, but still significantly better than PLRA's.

| Implementation | Gurobi | | | Flow ($w = 10$) | |
|---|---|---|---|---|---|
| Algorithm | PLRA | PM-Q | PM-E | PM-Q | PM-E |
| Wall Clock Time | 21.50s | 86.85s | 368.13s | 563.24s | 554.07s |
| Total CPU Time | 78.41s | 988.11s | 3212.18s | 562.77s | 553.59s |
| User CPU Time | 67.66s | 906.72s | 2957.54s | 562.68s | 553.42s |
| System CPU Time | 10.75s | 81.39s | 254.64s | 0.09s | 0.13s |
| Quality | 95% | 95% | 95% | 94% | 94% |
| PQuality | N/A | 606.55 | 305.41 | 603.34 | 303.68 |
| Maxprob ($\downarrow$) | 0.90 | 0.92 | 0.92 | 0.90 | 0.90 |
| AvgMaxp ($\downarrow$) | 0.90 | 0.87 | 0.87 | 0.86 | 0.86 |
| Support ($\uparrow$) | 3648 | 7122 | 7480 | 6250 | 6432 |
| Entropy ($\uparrow$) | 595.25 | 1589.95 | 1630.80 | 1575.86 | 1607.20 |
| L2Norm ($\downarrow$) | 47.83 | 42.49 | 42.45 | 42.21 | 42.18 |

Table 3: Comparison of different implementations on ICLR 2018. Downarrows ($\downarrow$) mean the lower the better and uparrows ($\uparrow$) mean the higher the better. Due to parallelization, Gurobi takes less wall clock time. However, the network-flow-based approximation uses fewer computation resources (total CPU time). The approximation has relatively accurate Quality, PQuality and randomness metrics.

**Comparing the running time of Gurobi and Algorithm 1.** As shown in Tables 2 and 3, Gurobi generally runs faster than Algorithm 1 in wall clock time. This is because it is a well-written commercial software that is able to parallelize over multiple cores on our server. However, the sum of user and system CPU time is smaller for Algorithm 1, which shows that Algorithm 1 uses fewer computation resources than Gurobi. Moreover, note that Gurobi takes significantly longer to solve PM-E than PM-Q, while Algorithm 1 is not affected. The reason for this is Gurobi only supports quadratic objective functions primitively. Therefore, to solve exponential objective functions, iterative approximation methods have to be used. In contrast, Algorithm 1 can support an arbitrary perturbation function without sacrificing running time.

**Comparing the solution qualities of Gurobi and Algorithm 1.** In the rows about Quality and PQuality in Tables 2 and 3, we can see that Algorithm 1 has almost the same Quality as Gurobi. The PQuality of Algorithm 1 is also close to Gurobi's. This shows that with $w = 10$, Algorithm 1 approximates PM in solution quality well, validating Theorem 1.

**Comparing the randomness metrics of Gurobi and Algorithm 1.** As shown in Table 2, for randomness metrics Maxprob, AvgMaxp, and L2Norm, the approximated solution by Algorithm 1 has a similar or identical performance to Gurobi on AAMAS 2015. For Support and Entropy, the approximated solution is significantly worse than Gurobi's, but it is still significantly better than PLRA's solution. As presented in Table 3, on ICLR 2018, the performances of Algorithm 1 and Gurobi on all randomness metrics are relatively close. This shows that Algorithm 1 mostly preserves PM's performance on randomness metrics. Although on discrete-valued datasets like AAMAS 2015, Support and Entropy can be affected, they are still better than PLRA's solution.

## A.2 Hyperparameter Tuning

In this section, we will introduce the way in which we do hyperparameter tuning for PM. We start with a lemma that shows the monotonicity of the solution quality of PM-Q with respect to $\beta$.

**Lemma A.1.** *For an input instance $(n_p, n_r, \ell_p, \ell_r, \mathbf{S})$, let the solution of PM with $f_1(x) = x - \beta_1 x^2$ be $\mathbf{x}_1$ and the solution of PM with $f_2(x) = x - \beta_2 x^2$ be $\mathbf{x}_2$ where $\beta_1, \beta_2 \in [0, 1]$. Then*

$$\beta_1 < \beta_2 \implies \text{Quality}(\mathbf{x}_1) \geq \text{Quality}(\mathbf{x}_2).$$

**Proof of Lemma A.1:** Let $\text{SQuality}(\mathbf{x}) = \sum_p \sum_r x_{p,r}^2 S_{p,r}$. Then

$$\text{PQuality}_{f_1}(\mathbf{x}) = \text{Quality}(\mathbf{x}) - \beta_1 \text{SQuality}(\mathbf{x}),$$
$$\text{PQuality}_{f_2}(\mathbf{x}) = \text{Quality}(\mathbf{x}) - \beta_2 \text{SQuality}(\mathbf{x}).$$

As $\mathbf{x}_1$ maximizes $\text{PQuality}_{f_1}(\mathbf{x})$ and $\mathbf{x}_2$ maximizes $\text{PQuality}_{f_2}(\mathbf{x})$, we have

$$\text{Quality}(\mathbf{x}_1) - \beta_1 \text{SQuality}(\mathbf{x}_1) \geq \text{Quality}(\mathbf{x}_2) - \beta_1 \text{SQuality}(\mathbf{x}_2), \tag{1}$$
$$\text{Quality}(\mathbf{x}_1) - \beta_2 \text{SQuality}(\mathbf{x}_1) \leq \text{Quality}(\mathbf{x}_2) - \beta_2 \text{SQuality}(\mathbf{x}_2). \tag{2}$$

Subtracting (2) from (1), and using $\beta_2 > \beta_1$, we have

$$(\beta_2 - \beta_1)\text{SQuality}(\mathbf{x}_1) \geq (\beta_2 - \beta_1)\text{SQuality}(\mathbf{x}_2) \implies \text{SQuality}(\mathbf{x}_1) \geq \text{SQuality}(\mathbf{x}_2).$$

Then, from (1), we know that

$$\text{Quality}(\mathbf{x}_1) - \text{Quality}(\mathbf{x}_2) \geq \beta_1(\text{SQuality}(\mathbf{x}_1) - \text{SQuality}(\mathbf{x}_2)) \geq 0.$$

Therefore, $\text{Quality}(\mathbf{x}_1) \geq \text{Quality}(\mathbf{x}_2)$. ∎

With Lemma A.1, for a minimum requirement of quality $\text{Quality}_M$, we can use binary search to find the largest $\beta$ for PM-Q such that PM-Q with $f(x) = x - \beta x^2$ gives an assignment with quality $\geq \text{Quality}_M$. Formally, consider the following Algorithm 2.

---

**Algorithm 2**: Hyperparameter Tuning for PM-Q

For a given minimum quality requirement $\text{Quality}_M$ and a slackness $\delta$:
    (1) Use binary search to find $Q_{\text{PLRA}}$, the smallest $Q$ with which PLRA gives a solution with quality $\geq \text{Quality}_M$.
    (2) Use binary search to find $\beta_{\max}$, the largest $\beta$ such that PM-Q with $Q = Q_{\text{PLRA}} + \delta$ and $f(x) = x - \beta^2$ gives a solution with quality $\geq \text{Quality}_M$.
    (3) Finalize the hyperparameters as $(Q, \beta) = (\min\{Q_{\text{PLRA}} + \delta, 1\}, \beta_{\max})$.

---

Algorithm 2 maximizes the parameter $\beta$ while ensuring the produced assignment with the set of hyperparameters $(Q, \beta)$ has quality at least $\text{Quality}_M$ and Maxprob at most $Q_{\text{PLRA}} + \delta$. Note that any solution with quality $\geq \text{Quality}_M$ has a Maxprob $\geq Q_{\text{PLRA}}$. Intuitively, Algorithm 2 is

trying to maximize the randomness while obtaining the minimum required quality and near-optimal $\mathrm{Maxprob}$ given the quality constraint.

Analogously, we use a similar method for tuning the hyperparameters in PM-E.

---

**Algorithm 3**: Hyperparameter Tuning for PM-E

For a given minimum quality requirement $\mathrm{Quality}_M$ and a slackness $\delta$:
(1) Use binary search to find $Q_{\mathrm{PLRA}}$, the smallest $Q$ with which PLRA gives a solution with quality $\geq \mathrm{Quality}_M$.
(2) Use binary search to find $\alpha_{\max}$, the largest $\alpha$ such that PM-E with $Q = Q_{\mathrm{PLRA}} + \delta$ and $f(x) = 1 - e^{-\alpha x}$ gives a solution with quality $\geq \mathrm{Quality}_M$.
(3) Finalize the hyperparameters as $(Q, \alpha) = (\min\{Q_{\mathrm{PLRA}} + \delta, 1\}, \alpha_{\max})$.

---

Although we similarly apply binary search to PM-E as we have done to PM-Q, the monotonicity condition of Lemma A.1 does not hold in general for PM-E. For instance, consider the following Example 1. If we run PM-E with different $\alpha$ and $Q = 1$, the solution quality is not monotonic with respect to $\alpha$. The results are shown in Fig. 5.

**Example 1.** *In this example, $n_p = n_r = 3$, $\ell_p = \ell_r = 1$ and*

$$\mathbf{S} = \begin{pmatrix} 0.4 & 0.0 & 0.6 \\ 0.8 & 0.6 & 0.0 \\ 0.8 & 0.6 & 1.0 \end{pmatrix}$$

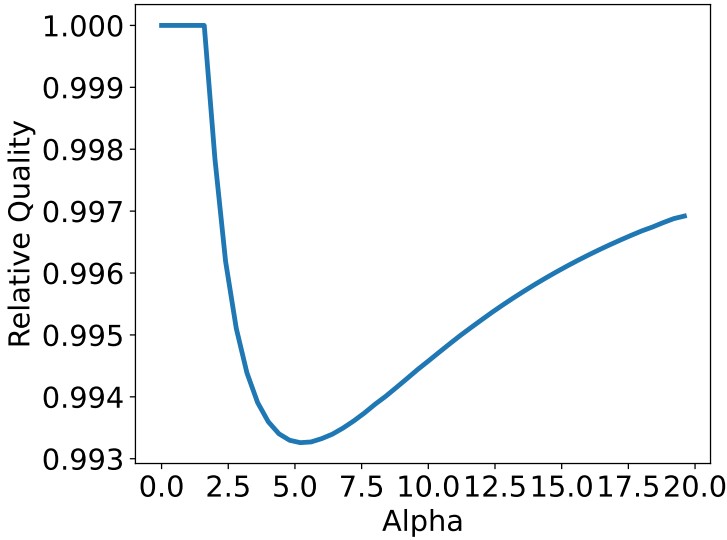

Figure 5: The quality of PM-E on Example 1 with $Q = 1$ and different $\alpha$. The figure shows that the solution quality of PM-E is not necessarily monotonic with respect to $\alpha$.

Nevertheless, such examples are rare in practice. We have examined thousands of examples to find Example 1. In fact, the monotonicity of solution quality with respect to $\alpha$ can still be observed empirically in the datasets we used. Therefore, for the purpose of our experiments in Section 6, we will still use Algorithm 3 to tune the hyperparameters of PM-E.

In Algorithms 2 and 3, there are still two parameters to be fixed, namely $\delta$ and $\mathrm{Quality}_M$. In Section 6, $\mathrm{Quality}_M$ was already specified as $\{80\%, 85\%, 90\%, 95\%, 98\%, 99\%, 99.5\%, 100\%\}$ of the maximum possible quality. It remains to specify $\delta$. For experiments in Section 6, we used $\delta = 0$ for AAMAS 2015 and $\delta = 0.02$ for ICLR 2018. In the rest of the section, we will show experiments about the performances of PM-Q and PM-E with different $\delta$ values to justify our choice. In practice, conference program chairs can simply choose a small value for the slackness, such as $\delta = 0.02$.

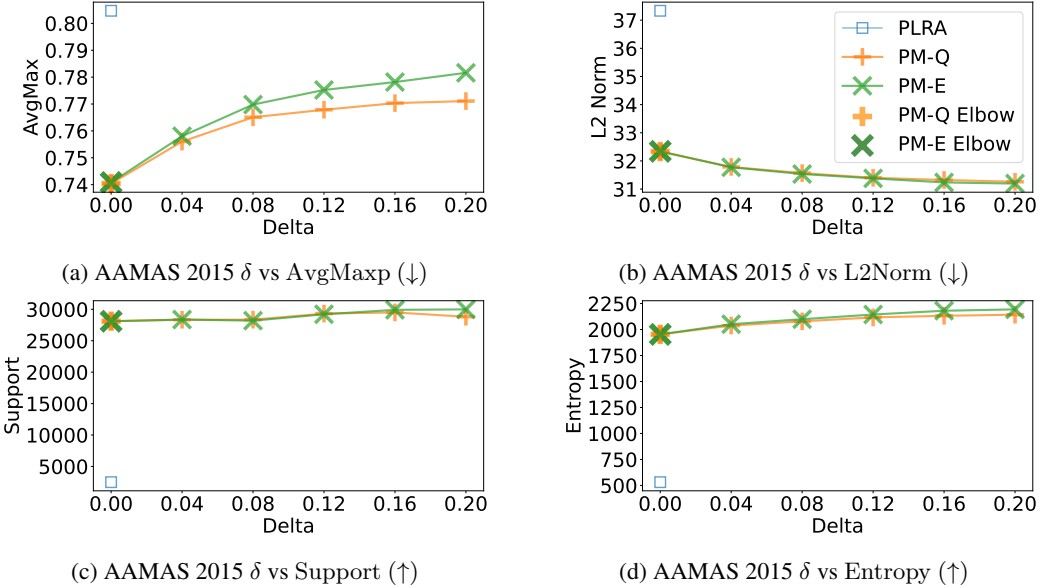

(a) AAMAS 2015 $\delta$ vs AvgMaxp ($\downarrow$)

(b) AAMAS 2015 $\delta$ vs L2Norm ($\downarrow$)

(c) AAMAS 2015 $\delta$ vs Support ($\uparrow$)

(d) AAMAS 2015 $\delta$ vs Entropy ($\uparrow$)

Figure 6: The performances of PM-Q and PM-E on AAMAS 2015 with different $\delta$. $\text{Quality}_M$ is set to 95% of the maximum possible quality. Downarrows ($\downarrow$) mean the lower the better and uparrows ($\uparrow$) mean the higher the better. The figure shows that the elbow points of both algorithms in all four metrics are at $\delta = 0$. Therefore, we should set $\delta = 0$ for AAMAS 2015.

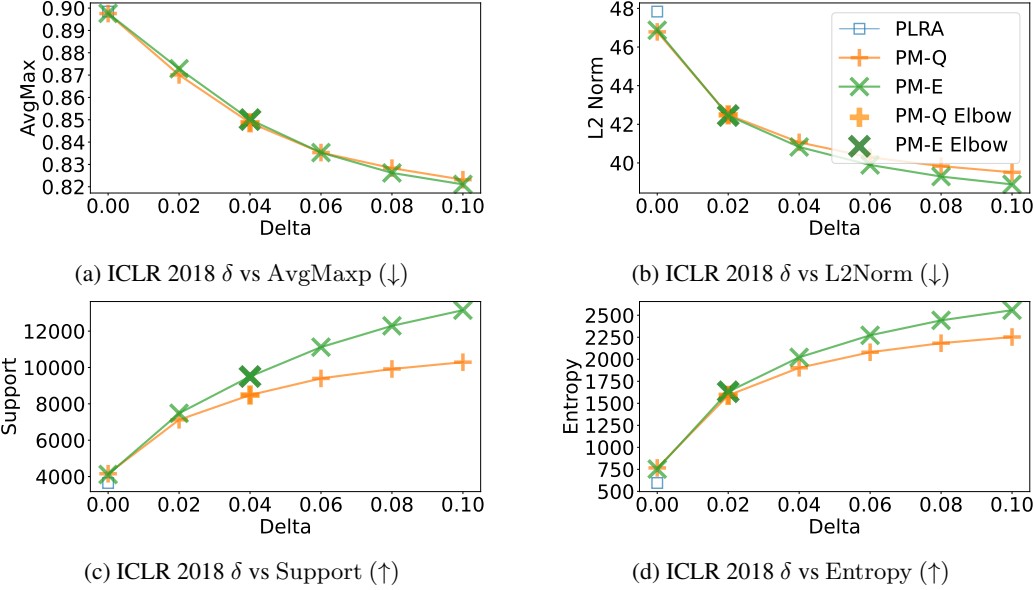

(a) ICLR 2018 $\delta$ vs AvgMaxp ($\downarrow$)

(b) ICLR 2018 $\delta$ vs L2Norm ($\downarrow$)

(c) ICLR 2018 $\delta$ vs Support ($\uparrow$)

(d) ICLR 2018 $\delta$ vs Entropy ($\uparrow$)

Figure 7: The performances of PM-Q and PM-E on ICLR 2018 with different $\delta$. $\text{Quality}_M$ is set to 95% of the maximum possible quality. Downarrows ($\downarrow$) mean the lower the better and uparrows ($\uparrow$) mean the higher the better. The figure shows that the elbow points of both algorithms lie in the range $\delta \in [0.02, 0.04]$. We have chosen $\delta = 0.02$ in this range.

As we increase $\delta$, we allow PM-Q and PM-E to find more random solutions (i.e., solutions that perform better on our randomness metrics) at a cost of larger $\text{Maxprob}$. To balance the gain and loss, we choose $\delta$ that maximizes a linear combination of them. We call these $\delta$ **elbow points**. Below, we show the choices of $\delta$ graphically in Figs. 6 and 7.

As shown in Figs. 6 and 7, on AAMAS 2015, the elbow points of both algorithms in all four metrics are at $\delta = 0$. Therefore, we should set $\delta = 0$ for it. On ICLR 2018, the elbow points lie in the range $\delta \in [0.02, 0.04]$. We have chosen $\delta = 0.02$ in this range for it.

## A.3 Experiments on More Datasets

In this subsection, we include additional experiment results on more 4 datasets. In particular, we include Preflib1 (00039-1 from [39], $n_p = 54, n_r = 31$), Preflib2 (00039-2 from [39], $n_p = 52, n_r = 24$), Preflib3 (00039-3 from [39], $n_p = 176, n_r = 146$) and AAMAS 2016 (00037-2 from [39], $n_p = 442, n_r = 161$). All of the four datasets are bidding data consisting of 4 discrete levels: "yes", "maybe", "no" and "conflict". We transform these 4 levels to similarities $1, \frac{1}{2}, \frac{1}{4}$ and $0$ as done in Section 6. The constraints are set as $\ell_p = 3, \ell_r = 6$ for Preflib1 and Preflib3. For Preflib2 and AAMAS 2016, $(\ell_p, \ell_r)$ are set as $(3, 7)$ and $(3, 12)$ respectively for feasibility.

We use the same experimental setup and hyperparameter setting as in Section 6. The results are shown in Fig. 8. The results of these additional datasets are similar to those on AAMAS 2015 shown in Figs. 2a and 3. In particular, both versions of PM achieve exactly the same performance with PLRA on $\mathrm{Maxprob}$ while improving significantly on the other randomness metrics. This shows that the empirical observations in Section 6 are not dataset-specific and generalize to other datasets.

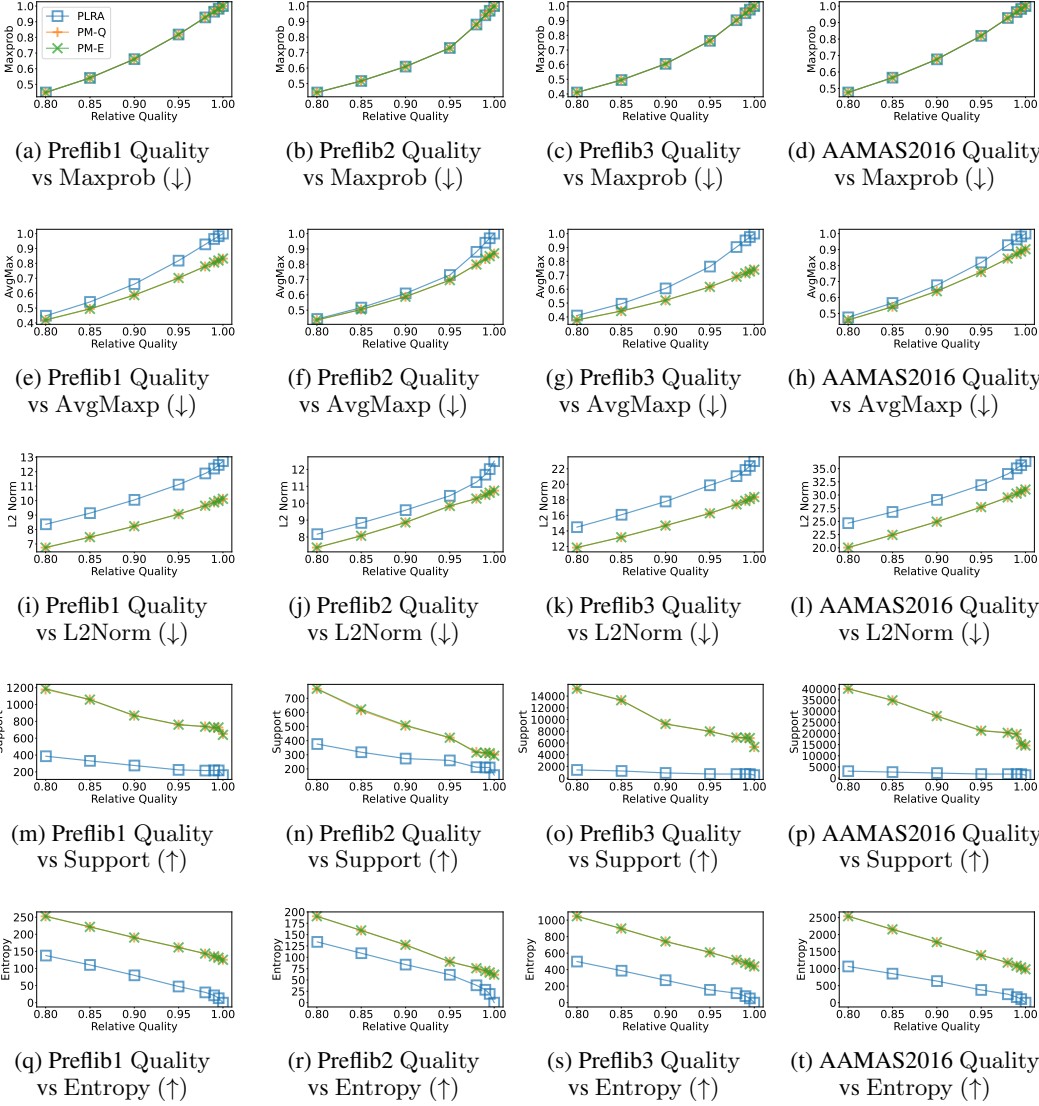

Figure 8: The trade-offs between quality and five randomness metrics on Preflib1, Preflib2, Preflib3 and AAMAS 2016. Downarrows ($\downarrow$) indicate that lower is better and uparrows ($\uparrow$) indicate that higher is better. The figure shows that PM incurs no increase in $\mathrm{Maxprob}$ while significantly improves over PLRA on all other four metrics. The results are similar to those on AAMAS 2015 shown in Section 6.

# B   Missing Proofs in Sections 4 and 5

## B.1   Proof of Theorem 1

**Theorem 1.** *Let $w \in \mathbb{N}^+$ be the precision in Algorithm 1, and* OPT *be the optimal perturbed quality.*

(a) *Algorithm 1 runs in $O(w \cdot \ell_p \cdot n_p{}^2 \cdot n_r)$ time.*

(b) *Algorithm 1 returns an assignment with perturbed quality* $\text{ALG} \geq \text{OPT} - f(\frac{1}{w}) \sum_{p,r} S_{p,r}$.

**Proof of Theorem 1:**   For (a), consider implementing the maximum cost flow using Edmonds & Karp's Algorithm B in [37]. In Edmonds & Karp's algorithm, we will compute the shortest path on the residual graph each time we find an augmenting path. The number of times we augment is bounded by $O(w \cdot \ell_p \cdot n_p)$. For the shortest path part, Edmonds & Karp's algorithm applies a node potential trick that ensures the edge weights are non-negative throughout the execution, so that we can use Dijkstra's algorithm for shortest path. Although for each paper $p$ and reviewer $r$, $v_p$ and $v_r$ are connected by $O(w)$ edges, only 2 of them need to be considered: the ones with the largest cost in both directions $v_p \rightarrow v_r$ and $v_r \rightarrow v_p$. Therefore, the shortest path can be computed in $O(n_p \cdot n_r)$ time. Step (6) of Algorithm 1 can be done in $O(n_p \cdot n_r)$ time by maintaining running totals for each reviewer and paper load. And thus Algorithm 1 runs in $O(w \cdot \ell_p \cdot n_p{}^2 \cdot n_r)$ time.

For (b), let $\mathbf{x}^*$ be the optimal assignment such that $\text{PQuality}_f(\mathbf{x}^*) = \text{OPT}$. Consider another assignment $\mathbf{x}'$, where $x'_{p,r} = \frac{1}{w} \lfloor w \cdot x^*_{p,r} \rfloor, \forall p \in \mathcal{P}, r \in \mathcal{R}$. Note that $\mathbf{x}'$ may not be a feasible assignment if $\sum_r x'_{p,r} \leq \ell_p$ for some paper $p$, but any feasible assignment $\mathbf{x}''$ where $x''_{p,r} \geq x'_{p,r}$ for all $(p,r)$ will have perturbed quality at least $\text{PQuality}_f(\mathbf{x}')$. Then $x^*_{p,r} - x'_{p,r} \in [0, \frac{1}{w}]$, and

$$\text{PQuality}_f(\mathbf{x}^*) - \text{PQuality}_f(\mathbf{x}') = \sum_{p,r} S_{p,r} \cdot [f(x^*_{p,r}) - f(x'_{p,r})] \leq f(\frac{1}{w}) \sum_{p,r} S_{p,r}. \quad (3)$$

On the other hand, $\mathbf{x}'$ corresponds to a feasible flow for the maximum-cost flow problem in Step (5) of Algorithm 1. Therefore, $\text{ALG} \geq \text{PQuality}_f(\mathbf{x}')$. Together with (3), we have $\text{ALG} \geq \text{OPT} - f(\frac{1}{w}) \sum_{p,r} S_{p,r}$. ∎

## B.2   Proof of Theorem 2

**Theorem 2.** *For an input instance $(n_p, n_r, \ell_p, \ell_r, \mathbf{S})$, where $\mathbf{S}$ is blockwise dominant with block identity $\mathbf{A} \in \mathbb{R}_{\geq 0}^{k \times k}$ and block sizes $\{p_1, \ldots, p_k\}, \{r_1, \ldots, r_k\}$, assume (i) $\{r_1, \ldots, r_k\}$ are not all equal, (ii) $p_i \cdot \ell_p \leq r_i \cdot \ell_r \; \forall i \in \{1, \ldots, k\}$ and (iii) $Q \cdot r_i \geq \ell_p \; \forall i \in \{1, \ldots, k\}$. Let $f$ be a strictly concave perturbation function and $f'(0) < \text{Dom}(\mathbf{A}) f'(1)$. PM with $f(x)$ as the perturbation function (weakly) dominates PLRA in quality and all randomness metrics. Formally,*

(a) $\forall \mathbf{x} \in \text{PM}(Q, f), \forall \mathbf{y} \in \text{PLRA}(Q)$,

$\text{Quality}(\mathbf{x}) \geq \text{Quality}(\mathbf{y}), \quad \text{Maxprob}(\mathbf{x}) \leq \text{Maxprob}(\mathbf{y}), \quad \text{AvgMaxp}(\mathbf{x}) \leq \text{AvgMaxp}(\mathbf{y}),$
$\text{L2Norm}(\mathbf{x}) \leq \text{L2Norm}(\mathbf{y}), \quad \text{Entropy}(\mathbf{x}) \geq \text{Entropy}(\mathbf{y}), \quad \text{Support}(\mathbf{x}) \geq \text{Support}(\mathbf{y}).$

(b) $\forall \mathbf{x} \in \text{PM}(Q, f), \exists \mathbf{y} \in \text{PLRA}(Q)$,

$\text{AvgMaxp}(\mathbf{x}) < \text{AvgMaxp}(\mathbf{y}), \quad \text{L2Norm}(\mathbf{x}) < \text{L2Norm}(\mathbf{y}), \quad \text{Entropy}(\mathbf{x}) > \text{Entropy}(\mathbf{y}).$

**Proof of Theorem 2:**   Label the first $p_1$ papers and $r_1$ reviewers as group 1, the next $p_2$ papers and $r_2$ reviewers as group 2, the next $p_3$ papers and $r_3$ reviewers as group 3 and so on.

For convenience of language, for an assignment $\mathbf{x}$, define **the total** Quality **of paper** $p$ to be $\sum_r x_{p,r} S_{p,r}$ and **the total** $\text{PQuality}_f$ **of paper** $p$ to be $\sum_r f(x_{p,r}) S_{p,r}$.

Define $\mathbf{x}^*$ to be the assignment such that for each paper $p$ and reviewer $r$, $x^*_{p,r} = \ell_p / r_i$ if they are in the same group $i$ and $x^*_{p,r} = 0$ otherwise. Note that $x^*$ is feasible, because $\ell_p / r_i \leq Q$ according to assumption (iii), each row of $\mathbf{x}^*$ sums to $\ell_p$ and each column of $x^*$ sums $\leq \ell_r$ according to assumption (ii). Intuitively, $x^*$ is a generalized version of Fig. 1a on blockwise dominant matrices.

**Claim B.2.1.** *For feasible assignment* $\mathbf{x}$, $\mathrm{Quality}(\mathbf{x}^*) \geq \mathrm{Quality}(\mathbf{x})$. *The equality holds if and only if* $\mathbf{x}$ *assigns paper $p$ to reviewer $r$ with positive probability only when they are in the same group.*

**Proof of Claim B.2.1:** First, $\mathrm{Quality}(\mathbf{x}^*) = \sum_{i=1}^k p_i r_i \cdot \ell_p / r_i \cdot A_{i,i} = \sum_{i=1}^k p_i \cdot \ell_p \cdot A_{i,i}$.

On the other hand, recall that $A_{i,i} > A_{i,j}, \forall j \neq i$. For a paper $p$ in group $i$, $p$ can only be assigned to $\ell_p$ reviewers, so the total similarity is at most $\ell_p \cdot A_{i,i}$. Therefore, $\mathrm{Quality}(\mathbf{x}) \leq \sum_{i=1}^k p_i \cdot \ell_p \cdot A_{i,i} = \mathrm{Quality}(\mathbf{x}^*)$. The equality holds if and only if each paper in group $i$ is assigned to $\ell_p$ reviewers with $A_{i,i}$ similarity, i.e., $\ell_p$ reviewers in the same group. ∎

As PLRA is maximizing Quality and $x^*$ is a feasible solution, we have the following corollary.

**Corollary B.2.1.** $\forall \mathbf{x} \in \mathrm{PLRA}(Q)$, $\mathbf{x}$ *assigns paper $p$ to reviewer $r$ with positive probability if and only if they are in the same group.*

Next, we consider the performance of PM.

**Claim B.2.2.** $\mathrm{PM}(Q, f) = \{\mathbf{x}^*\}$.

**Proof of Claim B.2.2:** First, $A_{i,i} \geq \mathrm{Dom}(\mathbf{A}) A_{i,j}, \forall j \neq i$ by the definition of $\mathrm{Dom}(\mathbf{A})$.

For a paper $p$ in group $i$, consider the maximum total $\mathrm{PQuality}_f$ of $p$. For some assignment $\mathbf{x}$, suppose $\mathbf{x}$ assigns $p$ to some reviewer $r$ in group $j \neq i$ with probability $\tau$. If we adjust $\mathbf{x}$ so that the probability $\tau$ is instead assigned to reviewers in group $i$, then the $\mathrm{PQuality}_f$ will first decrease by $f(\tau) \cdot A_{i,j} \leq \tau f'(0) A_{i,j}$ and then increase by at least $\tau f'(1) A_{i,i}$. As $f'(0) < \mathrm{Dom}(\mathbf{A}) f'(1)$,

$$\tau f'(1) A_{i,i} - \tau f'(0) A_{i,j} > \tau f'(1)(A_{i,i} - \mathrm{Dom}(\mathbf{A}) A_{i,j}) \geq 0.$$

This shows that the adjustment increases the total $\mathrm{PQuality}_f$ of $p$. Therefore, to maximize the total $\mathrm{PQuality}_f$ of $p$, $p$ should only be assigned to reviewers in group $i$. Let $p$ be assigned to the $j$-th reviewer in group $i$ with probability $\alpha_j$ ($\sum_j \alpha_j = \ell_p$), then the total $\mathrm{PQuality}_f$ of $p$ equals

$$\sum_{j=1}^{r_i} f(\alpha_j) \cdot A_{i,i}. \tag{4}$$

According to Jensen's inequality, only $\alpha_j = \frac{\ell_p}{r_i}, \forall j$ maximizes (4). This shows that, for every paper $p$, the corresponding row of $\mathbf{x}^*$ is the unique assignment that maximizes the total $\mathrm{PQuality}_f$ of $p$. And thus $\mathbf{x}^*$ is the unique solution that maximizes the overall $\mathrm{PQuality}_f$. ∎

Now we are ready to proceed to prove Theorem 2. Note that Claim B.2.2 shows that the only possible solution of PM is $\mathbf{x}^*$. For clearer presentation, let $\mathbf{x}^{(\mathrm{PM})} = \mathbf{x}^*$.

For Quality, Claim B.2.1 implies $\forall \mathbf{x}^{(\mathrm{PLRA})} \in \mathrm{PLRA}(Q)$, $\mathrm{Quality}(\mathbf{x}^{(\mathrm{PM})}) \geq \mathrm{Quality}(\mathbf{x}^{(\mathrm{PLRA})})$.

For Maxprob, compute that $\mathrm{Maxprob}(\mathbf{x}^{(\mathrm{PM})}) = \max_i \{\ell_p / r_i\}$. On the other hand, Corollary B.2.1 implies that $\forall \mathbf{x}^{(\mathrm{PLRA})} \in \mathrm{PLRA}(Q)$, $\mathrm{Maxprob}(\mathbf{x}^{(\mathrm{PLRA})}) \geq \max_i \{\ell_p / r_i\} = \mathrm{Maxprob}(\mathbf{x}^{(\mathrm{PM})})$.

For Support, compute that $\mathrm{Support}(\mathbf{x}^{(\mathrm{PM})}) = \sum_{i=1}^k p_i r_i$. On the other hand, Corollary B.2.1 implies that $\forall \mathbf{x}^{(\mathrm{PLRA})} \in \mathrm{PLRA}(Q)$, $\mathrm{Support}(\mathbf{x}^{(\mathrm{PLRA})}) \leq \sum_{i=1}^k p_i r_i = \mathrm{Support}(\mathbf{x}^{(\mathrm{PM})})$.

For the other metrics, $\mathrm{AvgMaxp}, \mathrm{Entropy}$ and $\mathrm{L2Norm}$, consider a paper $p$ in group $i$. By Corollary B.2.1, we know that PLRA only assigns $p$ to reviewers in group $i$. Let it be assigned to the $j$-th reviewer in group $i$ with probability $\alpha_j$ ($\sum_j \alpha_j = \ell_p$). Note that $\max_j \{\alpha_j\}$ is minimized by $\alpha_j = \frac{\ell_p}{r_i}, \forall j$, which is the corresponding row in $x^{(\mathrm{PM})}$ for each $p$. Meanwhile, according to Jensen's inequality, $\sum_j \alpha_j^2$ is minimized by $\alpha_j = \frac{\ell_p}{r_i}, \forall j$ and $\sum_j \alpha_j \ln(1/\alpha_j)$ is maximized by $\alpha_j = \frac{\ell_p}{r_i}, \forall j$. So $\forall \mathbf{x}^{(\mathrm{PLRA})} \in \mathrm{PLRA}(Q)$, $\mathrm{AvgMaxp}(\mathbf{x}^{(\mathrm{PM})}) \leq \mathrm{AvgMaxp}(\mathbf{x}^{(\mathrm{PLRA})})$, $\mathrm{L2Norm}(\mathbf{x}^{(\mathrm{PM})}) \leq \mathrm{L2Norm}(\mathbf{x}^{(\mathrm{PLRA})})$ and $\mathrm{Entropy}(\mathbf{x}^{(\mathrm{PM})}) \geq \mathrm{Entropy}(\mathbf{x}^{(\mathrm{PLRA})})$. Moreover, as $\{r_1, \ldots, r_k\}$ are not all the same, for some group $i$, $\frac{\ell_p}{r_i} < \max_j \{\frac{\ell_p}{r_j}\} \leq Q$. For this group, $\alpha_j = \frac{\ell_p}{r_i}, \forall j$ is not the only solution. Thus, $\exists \mathbf{x}^{(\mathrm{PLRA})} \in \mathrm{PLRA}(Q)$, $\mathrm{AvgMaxp}(\mathbf{x}^{(\mathrm{PM})}) < \mathrm{AvgMaxp}(\mathbf{x}^{(\mathrm{PLRA})})$, $\mathrm{L2Norm}(\mathbf{x}^{(\mathrm{PM})}) < \mathrm{L2Norm}(\mathbf{x}^{(\mathrm{PLRA})})$ and $\mathrm{Entropy}(\mathbf{x}^{(\mathrm{PM})}) > \mathrm{Entropy}(\mathbf{x}^{(\mathrm{PLRA})})$.

This concludes the proof of Theorem 2 ∎

## B.3   Proof of Theorem 3

**Theorem 3.** *For an input instance $(n_p, n_r, \ell_p, \ell_r, \mathbf{S})$, where each entry of $\mathbf{S}$ is i.i.d. sampled from $\{v_1, \ldots, v_k\}$ $(0 < v_1 < \cdots < v_k)$ uniformly, assume (i) $k \le \frac{1}{c} \cdot n_p$, (ii) $\ell_r \ge c \cdot \ln(n_r)$, (iii) $2 \cdot n_p \cdot \ell_p \le n_r \cdot \ell_r$ and (iv) $Q \cdot (n_r - 1) \ge \ell_p$. Let $f$ be a strictly concave perturbation function and $f'(0) < \frac{v_i}{v_{i-1}} f'(1), \forall i \in \{2, \ldots, k\}$. With probability $1 - e^{-\Omega(c)}$, PM with $f(x)$ as the perturbation function (weakly) dominates PLRA in quality and all randomness metrics. Formally,*

(a) $\forall \mathbf{x} \in \mathrm{PM}(Q, f)$, $\forall \mathbf{y} \in \mathrm{PLRA}(Q)$,

$\mathrm{Quality}(\mathbf{x}) \ge \mathrm{Quality}(\mathbf{y})$,    $\mathrm{Maxprob}(\mathbf{x}) \le \mathrm{Maxprob}(\mathbf{y})$,   $\mathrm{AvgMaxp}(\mathbf{x}) \le \mathrm{AvgMaxp}(\mathbf{y})$,
$\mathrm{L2Norm}(\mathbf{x}) \le \mathrm{L2Norm}(\mathbf{y})$,    $\mathrm{Entropy}(\mathbf{x}) \ge \mathrm{Entropy}(\mathbf{y})$,     $\mathrm{Support}(\mathbf{x}) \ge \mathrm{Support}(\mathbf{y})$.

(b) $\forall \mathbf{x} \in \mathrm{PM}(Q, f)$, $\exists \mathbf{y} \in \mathrm{PLRA}(Q)$,

$\mathrm{Support}(\mathbf{x}) > \mathrm{Support}(\mathbf{y})$,   $\mathrm{L2Norm}(\mathbf{x}) < \mathrm{L2Norm}(\mathbf{y})$,   $\mathrm{Entropy}(\mathbf{x}) > \mathrm{Entropy}(\mathbf{y})$.

**Proof of Theorem 3:**  To prove this statement, first consider the following 2 algorithms.

---

**Algorithm 4**: Greedy

For each paper $p$, let the reviewers sorted by decreasing similarity with $p$ be $\{r_1, r_2, \ldots, r_{n_r}\}$. Greedily assign $p$ to the first few reviewers as follows: assign $p$ to $r_1, r_2, \cdots, r_{\lfloor \ell_p/Q \rfloor}$ with probability $Q$, and assign $p$ to $r_{\lfloor \ell_p/Q \rfloor + 1}$ with probability $\ell_p - Q\lfloor \ell_p/Q \rfloor$.

---

**Algorithm 5**: Balanced Greedy

For each paper $p$, suppose $\forall i \in \{1, 2, \ldots, k\}$ the set of reviewers with similarity $v_i$ is $R_{p,i}$. Initialize remaining paper requirement $\ell_{\mathrm{remain}} = \ell_p$. For $i$ from $k$ to 1:

- If $\ell_{\mathrm{remain}} \ge Q \cdot |R_{p,i}|$, assign $p$ to each reviewer in $R_{p,i}$ with probability $Q$.
- Otherwise uniformly assign $p$ to each reviewer in $R_{p,i}$ with probability $\ell_{\mathrm{remain}}/|R_{p,i}|$.
- Update $\ell_{\mathrm{remain}}$ to the current remaining paper requirement $\max(0, \ell_{\mathrm{remain}} - Q|R_{p,i}|)$.

---

Note that Balanced Greedy is almost the same algorithm as Greedy, except that Balanced Greedy groups reviewers with the same similarity $v_i$ with $p$ together as $R_{p,i}$ and always assigns the same probability to them, while Greedy treats each reviewer individually.

These two Greedies do not always produce feasible assignments because they both consider each paper $p$ individually and do not take the constraint that each reviewer is assigned at most $\ell_r$ papers into account. Nevertheless, we will show that with high probability, they are feasible, and if this is the case, we can relate them to PM & PLRA and prove the Theorem 3.

At a high level, we prove the Theorem 3 with the following steps.

(1) Greedy and Balanced Greedy are feasible with probability $1 - e^{-\Omega(c)}$.

(2) When Greedy and Balanced Greedy are feasible:
- Greedy produces a possible solution from PLRA.
- PM does exactly the same with Balanced Greedy.

(3) Balanced Greedy dominates Greedy with probability $1 - e^{-\Omega(c)}$.

We will formalize and prove the above 3 steps below.

**Claim B.3.1.** *Greedy and Balanced Greedy are feasible with probability $1 - e^{-\Omega(c)}$.*

**Proof of Claim B.3.1:**  We use **Bernstein's inequality:** Suppose $x_1, x_2, \ldots, x_n$ are i.i.d. from a distribution with mean $\mu$, bounded support $[a, b]$, with variance $\sigma^2$. Then

$$\Pr[|\hat{\mu} - \mu| \ge t] \le 2 \exp\left(-\frac{n \cdot t^2}{2(\sigma^2 + (b-a)t)}\right) \text{ where } \hat{\mu} = \frac{1}{n}\sum_{i=1}^{n} x_i.$$

In Greedy and Balanced Greedy, the output $\mathbf{x}$ is a $n_p \times n_r$ matrix of random variables, where variables from different rows (papers) are independent as different papers are considered individually. These algorithms are feasible if and only if for each reviewer $r$, $\sum_p x_{p,r} \leq \ell_r$. Note that as the similarities are i.i.d. random, the reviewers are symmetric. Using Union Bound across reviewers,

$$\Pr[\text{ALG is infeasible}] \leq n_r \cdot \mathbf{Pr}\left[\sum_p x_{p,r_1} > \ell_r\right], \forall \text{ALG} \in \{\text{Greedy}, \text{Balanced Greedy}\}.$$

Moreover, the papers are also symmetric. So $\sum_p x_{p,r_1}$ is then a sum of $n_p$ i.i.d. random variables. Let their distribution be $D$. We will bound the probability using Bernstein's inequality. To do this, consider the properties of the distribution $D$. For both Greedy and Balanced Greedy,

- The expectation $\mu$ of $D$ should be $\ell_p/n_r$.
- Also, $D$ is supported on $[0, Q] \subseteq [0, 1]$.
- Therefore, $\sigma^2 = \mathbf{E}_{x\sim D}[x^2] - \mu^2 \leq \mathbf{E}_{x\sim D}[x^2] \leq \mathbf{E}_{x\sim D}[x] = \ell_p/n_r$.

Also, by assumption (iii), $2 \cdot n_p \cdot \ell_p \leq n_r \cdot \ell_r$, we then know $\ell_r/n_p \geq 2 \cdot \ell_p/n_r$. So that

$$\mathbf{Pr}\left[\sum_p x_{p,r_1} > \ell_r\right] = \mathbf{Pr}\left[|\hat{\mu} - \mu| > \frac{\ell_r}{n_p} - \mu\right] \leq \mathbf{Pr}\left[|\hat{\mu} - \mu| \geq \frac{\ell_r}{2n_p}\right].$$

Letting $t = \frac{\ell_r}{2n_p}$, we will use Bernstein's inequality on this formula. According to Bernstein's inequality,

$$\mathbf{Pr}\left[\sum_p x_{p,r_1} > \ell_r\right] \leq 2\exp\left(-\frac{n_p \cdot (\ell_r/2n_p)^2}{2(\ell_p/n_r + (1-0)(\ell_r/2n_p))}\right) \leq 2\exp\left(-\frac{1}{8}\ell_r\right).$$

Then, by assumption (ii), $\ell_r \geq c \cdot \ln(n_r)$, and we will see $\forall \text{ALG} \in \{\text{Greedy}, \text{Balanced Greedy}\}$,

$$\Pr[\text{ALG is infeasible}] \leq 2n_r \cdot e^{-\frac{1}{8}\ell_r} = 2e^{\left(1-\frac{1}{8}c\right)\ln(n_r)} = e^{-\Omega(c)}.$$

This concludes the proof of Claim B.3.1 ∎

For convenience of languague, for an assignment $\mathbf{x}$, define **the total** Quality **of paper** $p$ to be $\sum_r x_{p,r}S_{p,r}$ and **the total** PQuality$_f$ **of paper** $p$ to be $\sum_r f(x_{p,r})S_{p,r}$.

Let $\mathbf{x}^{(b)} = \text{Balanced Greedy}(Q)$, $\mathbf{x}^{(g)} = \text{Greedy}(Q)$. For paper $p$, both Balanced Greedy and Greedy maximize the total Quality of $p$, so Quality$(\mathbf{x}^{(b)}) = $ Quality$(\mathbf{x}^{(g)})$. Moreover, this property of the Greedies also gives us the following Claim B.3.2.

**Claim B.3.2.** *For any feasible assignment* $\mathbf{x}$, Quality$(\mathbf{x}^{(b)}) \geq$ Quality$(\mathbf{x})$. *The equality holds if and only if for each paper* $p$, $\mathbf{x}$ *maximizes the total* Quality *of* $p$.

And as PLRA is maximizing Quality, we further get the following Corollary B.3.1.

**Corollary B.3.1.** *If* $\mathbf{x}^{(b)}$ *and* $\mathbf{x}^{(g)}$ *are feasible, then* $\mathbf{x}^{(b)}, \mathbf{x}^{(g)} \in \text{PLRA}(Q)$ *and* $\forall \mathbf{x} \in \text{PLRA}(Q)$, $\mathbf{x}$ *maximizes the total* Quality *of* $p$.

Next, we consider the performance of PM.

**Claim B.3.3.** *For each paper* $p$, $\mathbf{x}^{(b)}$ *uniquely maximizes the total* PQuality$_f$ *of* $p$.

**Proof of Claim B.3.3:** For each paper $p$, recall that $\forall i \in \{1, 2, \ldots, k\}$, the set of reviewers with similarity $v_i$ to $p$ is $R_{p,i}$. By the execution of Balanced Greedy, $\mathbf{x}^{(b)}$ assigns $p$ to every reviewer in the $i$-th set, $R_{p,i}$, with the same probability. Let this probability be $A_{p,i}$. Then, there exists a $t \in \{1, 2, \ldots, k\}$, such that $A_{p,t+1} = \cdots = A_{p,k} = Q$, $A_{p,t} < Q$ and $A_{p,t-1} = \cdots = A_{p,1} = 0$.

Suppose that for an assignment $\mathbf{x}$, there are both a reviewer $r_{\text{high}} \in R_{p,i}$ such that $x_{p,r_{\text{high}}} < Q$, and another reviewer $r_{\text{low}} \in R_{p,j}$ $(j < i)$ such that $x_{p,r_{\text{low}}} > 0$. Let $\delta = \min\{Q - x_{p,r_{\text{high}}}, x_{p,r_{\text{low}}}\}$. Consider adjusting $\mathbf{x}$ by decreasing $x_{p,r_{\text{low}}}$ by $\delta$ and increasing $x_{p,r_{\text{high}}}$ by $\delta$. The total PQuality$_f$ of

$p$ will first decrease by at most $\delta f'(0)v_j$ and then increase by at least $\delta f'(1)v_i$. The net increase in the total $\mathrm{PQuality}_f$ of $p$ will be

$$\geq \delta(f'(1)v_i - f'(0)v_j) \geq \delta(f'(1)v_i - f'(0)v_{i-1}) > 0.$$

This shows that the adjustment increases the total $\mathrm{PQuality}_f$ of $p$, so $\mathbf{x}$ does not maximize it. Therefore, for an assignment $\mathbf{x}$ to maximize the total $\mathrm{PQuality}_f$ of $p$, $\mathbf{x}$ must assign $p$ to every reviewer in $R_{p,t+1}, \ldots, R_{p,k}$ with probability $Q$, and assign $p$ to every reviewer in $R_{p,t-1}, \ldots, R_{p,1}$ with probability $0$. It remains to consider the assignment to group $R_{p,t}$.

Write the total $\mathrm{PQuality}_f$ of $p$ in $\mathbf{x}$ as

$$\sum_{i=t+1}^{k} |R_{p,i}| \cdot f(1)v_i + \sum_{r \in R_{p,t}} f(x_{p,r})v_t$$

For fixed $p$, the first summation is constant. To maximize the second summation, according the Jensen's inequality, $x_{p,r}, \forall r \in R_{p,t}$ must be the same, which is exactly $\mathbf{x}^{(b)}$.

This concludes the proof of Claim B.3.3. ∎

Claim B.3.3 gives us the following Corollary B.3.2.

**Corollary B.3.2.** *If $\mathbf{x}^{(b)}$ is feasible,* $\mathrm{PM}(Q, f) = \{\mathbf{x}^{(b)}\}$.

Now we are ready to proceed to prove Theorem 3. We first prove Theorem 3 (a).

For Quality, Claim B.3.2 implies that $\forall \mathbf{x} \in \mathrm{PLRA}(Q)$, $\mathrm{Quality}(\mathbf{x}^{(b)}) \geq \mathrm{Quality}(\mathbf{x})$.

For the randomness metrics, consider each paper $p$ and again let the set of reviewers with similarity $v_i$ to $p$ be $R_{p,i}$. Like in the proof of Claim B.3.3, by the execution of Balanced Greedy, $\mathbf{x}^{(b)}$ assigns $p$ to every reviewer in $R_{p,i}$ with the same probability $A_{p,i}$, and there exists a $t \in \{1, 2, \ldots, k\}$, such that $A_{p,t+1} = \cdots = A_{p,k} = Q$, $A_{p,t} < Q$ and $A_{p,t-1} = \cdots = A_{p,1} = 0$.

By Corollary B.3.1, $\forall \mathbf{x} \in \mathrm{PLRA}(Q)$, $\mathbf{x}$ must assign $p$ to all reviewers in $R_{p,t+1}, \ldots, R_{p,k}$ with probability $Q$ and assign $p$ to all reviewers in $R_{p,1}, \ldots, R_{p,t-1}$ with probability $0$. Among all such assignments, with an argument using Jensen's inequality or simple categorical discussion, we will see that $\mathbf{x} = \mathbf{x}^{(b)}$ maximizes $\sum_r x_{p,r} \ln(1/x_{p,r})$, $\sum_r \mathbb{I}[x_{p,r} > 0]$ and minimizes $\max_r\{x_{p,r}\}$, $\sum_r x_{p,r}^2$.

This shows that Theorem 3 (a) holds.

To show Theorem 3 (b), we will first prove the following Claim B.3.4.

**Claim B.3.4.** *With probability $1 - e^{-\Omega(c)}$:*

$$\mathrm{Support}(\mathbf{x}^{(b)}) > \mathrm{Support}(\mathbf{x}^{(g)}), \mathrm{L2Norm}(\mathbf{x}^{(b)}) < \mathrm{L2Norm}(\mathbf{x}^{(g)}), \mathrm{Entropy}(\mathbf{x}^{(b)}) > \mathrm{Entropy}(\mathbf{x}^{(g)}).$$

**Proof of Claim B.3.4:** Recall in Greedy, for each paper $p$, the sorted reviewer list by decreasing similarity with $p$ is $\{r_1, r_2, \ldots, r_{n_r}\}$. By assumption (iv), $Q \cdot (n_r - 1) \geq \ell_p$, so $\lceil \ell_p/Q \rceil + 1 \leq n_r$.

Suppose for some paper $p$ and $i \in \{1, \ldots, k\}$, $S_{p,r_{\lceil \ell_p/Q \rceil}} = S_{p,r_{\lceil \ell_p/Q \rceil+1}} = v_i$. Then, as by the execution of Greedy, $x_{p,r_{\lceil \ell_p/Q \rceil}}^{(g)} > 0$ and $x_{p,r_{\lceil \ell_p/Q \rceil+1}}^{(g)} = 0$. Let $\{r_{\mathrm{left}}, \ldots, r_{\mathrm{right}}\}$ be the set of reviewers with similarity $v_i$ to $p$, where $\mathrm{left} \leq \lceil \ell_p/Q \rceil < \lceil \ell_p/Q \rceil + 1 \leq \mathrm{right}$. Then

$$\mathbf{x}_{p,r_j}^{(b)} = \begin{cases} Q & (j \leq \mathrm{left} - 1) \\ \frac{\ell_p - Q(\mathrm{left}-1)}{\mathrm{right}-\mathrm{left}+1} & (\mathrm{left} \leq j \leq \mathrm{right}) \\ 0 & (j \geq \mathrm{right} + 1) \end{cases},$$

$$\mathbf{x}_{p,r_j}^{(g)} = \begin{cases} Q & (j \leq \lceil \ell_p/Q \rceil - 1) \\ \ell_p - Q(\lceil \ell_p/Q \rceil - 1) & (j = \lceil \ell_p/Q \rceil) \\ 0 & (j \geq \lceil \ell_p/Q \rceil + 1) \end{cases}.$$

Note that $\mathbf{x}_{p,r_j}^{(b)} = \mathbf{x}_{p,r_j}^{(g)}$ for any $j \leq \mathrm{left} - 1$ and any $j \geq \mathrm{right} + 1$. For $\mathrm{left} \leq j \leq \mathrm{right}$, $\mathbf{x}_{p,r_j}^{(b)}$ are all equal, but as $x_{p,r_{\lceil \ell_p/Q \rceil+1}}^{(g)} = 0$, $\mathbf{x}_{p,r_j}^{(g)}$ are not all equal. So $\mathrm{Support}(\mathbf{x}^{(b)}) > \mathrm{Support}(\mathbf{x}^{(g)})$, and by Jensen's inequality, $\mathrm{L2Norm}(\mathbf{x}^{(b)}) < \mathrm{L2Norm}(\mathbf{x}^{(g)})$, $\mathrm{Entropy}(\mathbf{x}^{(b)}) > \mathrm{Entropy}(\mathbf{x}^{(g)})$.

Therefore, it remains to show that with probability $1 - e^{-\Omega(c)}$, for some paper $p$, $S_{p,r_{\lceil \ell_p/Q \rceil}} = S_{p,r_{\lceil \ell_p/Q \rceil+1}}$. For a fixed $p$, denote the event that $S_{p,r_{\lceil \ell_p/Q \rceil}} = S_{p,r_{\lceil \ell_p/Q \rceil+1}}$ as $E_p$. Recall that entries in $\mathbf{S}$ are i.i.d. and uniformly chosen from $\{v_1, \ldots, v_k\}$. Consider fixing $S_{p,1}, \ldots, S_{p,n_r-1}$, and let the $\lceil \ell_p/Q \rceil$-th largest number in this set be $\alpha$. Then $\Pr[S_{p,n_r} = \alpha] \geq \frac{1}{k}$. As $S_{p,n_r} = \alpha$ implies $E_p$, $\Pr[E_p = 1] \geq \frac{1}{k}$. Therefore, according to assumption (i), $k \leq \frac{1}{c} \cdot n_p$, we know that

$$\Pr[\exists p, E_p = 1] \geq 1 - \left(1 - \frac{1}{k}\right)^{n_p} \geq 1 - \left(1 - \frac{1}{k}\right)^{ck} \geq 1 - e^{-c}.$$

This concludes the proof of Claim B.3.4. ∎

Theorem 3 (b) is implied by Claim B.3.1, Corollary B.3.1, Corollary B.3.2 and Claim B.3.4. ∎

## C  Discussions

### C.1  Directly Optimizing Specific Randomness Metrics with PM

While PM provides one way to introduce randomness into the paper assignment, one natural alternative approach is to simply maximize one specific randomness metric, subject to a constraint on the minimum solution quality. In this section, we show that a slight modification to PM is general enough to capture such approaches.

Specifically, suppose we want to maximize a concave randomness metric $\text{RM} : [0,1]^{n_p \times n_r} \to \mathbb{R}$ over the set of feasible assignments $S_{\text{feasible}} = \{\mathbf{x} \in [0,1]^{n_p \times n_r} \mid \sum_r x_{p,r} = \ell_p, \forall p \text{ and } \sum_p x_{p,r} \leq \ell_r, \forall r\}$ subject to a minimum requirement of the solution quality $\text{Quality}(\mathbf{x}) \geq \tau \cdot \text{Quality}_{\text{OPT}}$, where $\text{Quality}_{\text{OPT}} = \max_{\mathbf{x} \in S_{\text{feasible}}} \text{Quality}(\mathbf{x})$ and $\tau \in [0,1]$. The problem can be formulated as

$$\begin{array}{ll} \text{Maximize} & \text{RM}(\mathbf{x}) \\ \text{Subject to} & \text{Quality}(\mathbf{x}) \geq \tau \cdot \text{Quality}_{\text{OPT}}, \\ & \mathbf{x} \in S_{\text{feasible}}. \end{array} \qquad (5)$$

We then consider a slight generalization of PM that allows different perturbation functions for each reviewer-paper pair. That is, we change the definition of $\text{PQuality}_f$ (the objective of PM) in Definition 4.2 to

$$\text{PQuality}_f(\mathbf{x}) = \sum_p \sum_r S_{p,r} \cdot f_{p,r}(x_{p,r}).$$

Then we have the following result:

**Theorem 4.** *If $f_{p,r}(x)$ is (i) $x + \frac{\lambda}{S_{p,r}}\mathbb{I}[x > 0]$, (ii) $x - \frac{\lambda}{S_{p,r}}x\ln(x)$, or (iii) $x - \frac{\lambda}{S_{p,r}}x^2$, then PM achieves the optimal trade-offs between $\text{Quality}(\mathbf{x})$ and (i) $\text{Support}(\mathbf{x})$, (ii) $\text{Entropy}(\mathbf{x})$, or (iii) $\text{L2Norm}^2(\mathbf{x})$ respectively with different values of $\lambda \in [0, +\infty)$.*

In fact, the proof of Theorem 4 also shows that PM-Q can be viewed as a algorithm that achieves the optimal trade-off between $\text{Quality}(\mathbf{x})$ and another randomness metric $(\sum_{p,r} S_{p,r} x_{p,r}^2)$, a similarity-weighted version of squared L2 norm of $\mathbf{x}$. Next, we will proceed to prove Theorem 4.

**Proof of Theorem 4:**  When $\tau \in [0,1]$, the maximization problem (5) always has at least one feasible solution. Let the optimal value of (5) be $\text{RM}^*(\tau)$. $\text{RM}^*(\tau)$ describes the optimal trade-off between $\text{Quality}(\mathbf{x})$ and $\text{RM}(\mathbf{x})$. We can show the following property of $\text{RM}^*(\tau)$.

**Lemma C.1.** *$\text{RM}^*(\tau)$ is a concave and non-increasing function of $\tau \in [0,1]$.*

**Proof of Lemma C.1:**  Let $\mathbf{x}_1$ be the optimal solution of (5) when $\tau = \tau_1$ and $\mathbf{x}_2$ be the optimal solution of (5) when $\tau = \tau_2$ where $0 \leq \tau_1 < \tau_2 \leq 1$. Then, we have $\text{Quality}(\mathbf{x}_2) \geq \tau_2 \cdot \text{Quality}_{\text{OPT}} > \tau_1 \cdot \text{Quality}_{\text{OPT}}$ and $\mathbf{x}_2$ is a feasible solution of of (5) when $\tau = \tau_1$. Therefore, $\text{RM}(\mathbf{x}_1) \geq \text{RM}(\mathbf{x}_2)$, i.e., $\text{RM}^*(\tau_1) \geq \text{RM}^*(\tau_2)$. This shows that $\text{RM}^*(\tau)$ is a non-increasing function of $\tau$.

Moreover, let $\mathbf{x}_m = \frac{1}{2}(\mathbf{x}_1 + \mathbf{x}_2)$. Then, $\text{Quality}(\mathbf{x}_m) \geq \frac{1}{2}(\tau_1 + \tau_2) \cdot \text{Quality}_{\text{OPT}}$ and $\mathbf{x}_m \in S_{\text{feasible}}$. Thus $\mathbf{x}_m$ is a feasible solution of of (5) when $\tau = \frac{1}{2}(\tau_1 + \tau_2)$. Since RM is a concave function,

$\mathrm{RM}^*(\frac{1}{2}(\tau_1 + \tau_2)) \geq \mathrm{RM}(\mathbf{x}_m) \geq \frac{1}{2}(\mathrm{RM}(\mathbf{x}_1) + \mathrm{RM}(\mathbf{x}_2)) = \frac{1}{2}(\mathrm{RM}^*(\tau_1) + \mathrm{RM}^*(\tau_2))$. This shows that $\mathrm{RM}^*(\tau)$ is a concave function of $\tau$. ∎

Now, for $\lambda \geq 0$, consider another optimization program as follows:

$$\begin{aligned} \text{Maximize} \quad & \text{Quality}(\mathbf{x}) + \lambda \cdot \mathrm{RM}(\mathbf{x}) \\ \text{Subject to} \quad & \mathbf{x} \in S_{\text{feasible}}. \end{aligned} \tag{6}$$

Let the solution of (6) be $x_\lambda^*$. Lemma C.1 gives us the following corollary.

**Corollary C.1.1.**

$$\{(\tau, \mathrm{RM}^*(\tau)) \mid \tau \in (0, 1]\} = \{(\text{Quality}(\mathbf{x}_\lambda^*)/\text{Quality}_{\text{OPT}}, \mathrm{RM}(\mathbf{x}_\lambda^*)) \mid \lambda \in [0, +\infty)\}.$$

Plugging in different randomness metrics $\mathrm{RM}(\mathbf{x})$ concludes the proof of Theorem 4. ∎

## C.2 Incorporating Various Constraints in PM

In some of the currently-deployed conference review systems, there are various additional constraints on the assignment of papers to reviewers. We will discuss in this section how to incorporate some of these constraints in PM. Specifically, we consider the following constraints from [20]:

(1) **Seniority:** Each paper is assigned to $\geq 1$ senior reviewer.

(2) **Geographic diversity:** No 2 reviewers assigned to the same paper belong to the same region.

**Seniority.** Let the set of senior reviewers be $S_{\text{senior}}$. We can incorporate the seniority constraint in PM by modifying PM to the following optimization program.

$$\begin{aligned} \text{Maximize} \quad & \text{PQuality}_f(\mathbf{x}) \\ \text{Subject to} \quad & \sum_r x_{p,r} = \ell_p & \forall p \in \mathcal{P}, \\ & \sum_{r \in S_{\text{senior}}} x_{p,r} \geq 1 & \forall p \in \mathcal{P}, \\ & \sum_p x_{p,r} \leq \ell_r & \forall r \in \mathcal{R}, \\ & 0 \leq x_{p,r} \leq Q & \forall p \in \mathcal{P}, r \in \mathcal{R}. \end{aligned}$$

With a slightly different sampling algorithm, we can show that the obtained randomized assignment can be realized by a distribution of deterministic assignments that satisfies the seniority constraint [22].

**Geographic diversity.** Let the set of regions be $S_{\text{region}}$. We can incorporate the geographic diversity constraint in PM by modifying PM to the following optimization program.

$$\begin{aligned} \text{Maximize} \quad & \text{PQuality}_f(\mathbf{x}) \\ \text{Subject to} \quad & \sum_r x_{p,r} = \ell_p & \forall p \in \mathcal{P}, \\ & \sum_{r \text{ belongs to } g} x_{p,r} \leq 1 & \forall p \in \mathcal{P}, g \in S_{\text{region}}, \\ & \sum_p x_{p,r} \leq \ell_r & \forall r \in \mathcal{R}, \\ & 0 \leq x_{p,r} \leq Q & \forall p \in \mathcal{P}, r \in \mathcal{R}. \end{aligned}$$

The obtained assignment can also be realized by a distribution of deterministic assignments that satisfies the geographic diversity constraint with a slightly different sampling algorithm [22].

Apart from the constraints mentioned above, there are also some other common constraints that cannot be easily incorporated in PM [20]. We leave incorporating them for future work.

