# OpenReview forum: "A One-Size-Fits-All Approach to Improving Randomness in Paper Assignment"
_NeurIPS.cc/2023/Conference — NeurIPS 2023 spotlight_

### Official Review · Reviewer_625T · 2023-07-03

**Soundness:** 3 good
**Presentation:** 3 good
**Contribution:** 2 fair
**Rating:** 6
**Confidence:** 3

**Summary:**

This paper focuses on randomized paper assignments in the reviewer-paper matching problem. First, the authors state the motivation and emphasize the importance of randomness in paper assignments. Then, they point out the weakness of the existing method, PLRA, and propose additional metrics for evaluating the performance of randomized paper assignments. After that, the authors design the PM-E and PM-Q algorithms with different perturbation functions to solve the proposed problem and provide theoretical analysis to prove the effectiveness of the proposed algorithms. Finally, various experiments have been conducted, showing that PM-E and PM-Q perform well on real-world datasets. In conclusion, this paper proposes practical and universal algorithms, PM-E and PM-Q, with the aim of mitigating malicious behaviors, facilitating assignment evaluation, and increasing reviewer diversity and anonymity. However, the theoretical analysis for the correctness of the approximate algorithm is inadequate.

**Strengths:**

1. This paper is well-written and easy to understand.
2. Related works are described in detail and the limitations of existing methods are well illustrated by a simple example.
3. The theoretical analyses regarding how the proposed PM algorithm outperforms PLRA are sufficient.
4. Various experiments show that the proposed algorithm has better performance than existing randomzied paper assignment methods.

**Weaknesses:**

1. Algorithm 1 prposes a network-flow-based approximation of PM and use experiments to show its pratical performance, but the theoretical analysis of how this algorithm approximate solves PM problem and approximation ratio is missing.
2. The motivations listed in Introduction mainly talk about randomness in paper assignments, but randomized paper assignment problem has been proposed in PLRA.

**Questions:**

1. Why is the additional metric valid? Could the authors provide some analysis or examples to briefly explain it?
2. Figure 1 presents the ideal assignment, but it assigns more papers to each reviewer. Therefore, is it possible that in a scenario where the quality provided by PLRA is similar to PM-E and PM-Q, PLRA assigns fewer articles to each reviewer resulting in a lower workload for each reviewer?

**Limitations:**

This paper acknowledges the limitation that deploying a randomized assignment may have a negative impact on assignment quality compared to deterministic assignments. Additionally, the authors propose a reasonable explanation that their work allows conferences to choose the level of randomization to address this limitation.

---

> ### Author Rebuttal · Authors · 2023-08-06
>
> Thank you for spending time understanding our work and providing insightful comments. We are happy to see that you find our work well-written, practical, and universal. Below we respond to the questions and weaknesses raised in your review, hoping to address your concerns about our paper.
>
> **Regarding the approximation ratio of Algorithm 1.**
>
> We do have a theoretical guarantee for Algorithm 1 that is not included in the paper. Let the optimal perturbed quality be $\mathrm{OPT}$. Algorithm 1 produces a solution of perturbed quality $\mathrm{ALG}$, where $0\leq \mathrm{OPT}-\mathrm{ALG}\leq f(\frac{1}{w})\sum_{p,r}S_{p,r}$. For a fixed input instance and a fixed perturbation function $f$, $\mathrm{ALG}$ approaches $\mathrm{OPT}$ as $w\to +\infty$. This formalizes the intuition that as we increase the precision $w$, the approximation becomes more accurate. We are happy to add this guarantee to the appendix in the next version.
>
> **Regarding the motivations.**
>
> Indeed, the randomized paper assignment problem has been proposed in [22]. However, [22] focused on one metric of randomness: the maximum assignment probability ($\mathrm{Maxprob}$). As a result, it is not clear how well [22] aligns with different motivations for randomization other than robustness to malicious behavior (our Motivations 2, 3, and 4 in Section 1). These additional motivations for randomization were not identified in [22]. In fact, these are important reasons why we should care about the general notion of randomness in paper assignments, rather than about optimizing a specific metric (like $\mathrm{Maxprob}$ in [22]). In practice, conference program chairs likely have some interest in all of these motivations, and want to deploy an assignment that can satisfy all of them as much as possible. Also, even for motivation 1, the metric used in [22] only corresponds to one specific malicious behavior model and our proposed random metric broadens the scope of malicious behavior models as we explain in detail.
>
> **Regarding the additional metrics.**
>
> The additional metrics are valid because of the following reasons:
>
> - Average maximum probability ($\mathrm{AvgMaxp}$): $\mathrm{AvgMaxp}$ corresponds to robustness to malicious behavior (Motivation 1 in Section 1). Depending on the model of malicious behavior, different metrics can fit into the motivation. The vanilla randomness metric $\mathrm{Maxprob}(\mathbf x)$ proposed in [22] corresponds to a model where the malicious reviewer-paper pair is a single worst-case pair chosen by Nature. However, if a particular paper $p$ is targeted, then in the worst case, the malicious reviewer targeting $p$ succeeds with probability $\max_{r}\lbrace x_{p,r}\rbrace$. Our new metric $\mathrm{AvgMaxp}(\mathbf x)$ calculates the average of these probabilities, corresponding to the model where a random paper $p$ is targeted with a worst-case malicious reviewer.
> - Support size ($\mathrm{Support}$): $\mathrm{Support}$ corresponds to evaluation of alternative assignments (Motivation 2 in Section 1). The techniques for off-policy evaluation discussed in [21] can be utilized to estimate the review quality of another non-deployed assignment. Crucially, the precision of the estimation depends on the number of "positivity violations'' [21] between the deployed assignment $\mathbf x$ and the non-deployed assignment $\mathbf y$ of interest with positivity violation defined as an entry $(i,j)$ where $x_{i,j}=0$ but $y_{i,j}\ne 0$. Since a large number of alternative assignments $\mathbf y$ may be of interest, to minimize the number of positivity violations, we want to minimize the number of entries in $\mathbf x$ set to  $0$. This corresponds to maximizing $\mathrm{Support}(\mathbf x)$.
> - Entropy ($\mathrm{Entropy}$): $\mathrm{Entropy}$ corresponds to the general idea of maximizing randomness. In information theory, entropy characterizes the uncertainty of a random variable. By maximizing entropy, we maximize the uncertainty of our assignment.
> - L2 Norm ($\mathrm{L2Norm}$): $\mathrm{L2Norm}$ also corresponds to the general idea of maximizing randomness. Note that for a feasible assignment $\mathrm x$, $\sum_{p,r}x_{p,r}=n_p\ell_p$ must hold. When the assignment is deterministic, $\sum_{p,r}x_{p,r}^2=n_p\ell_p$, this is the largest $\mathrm{L2Norm}$ a feasible assignment can get. Conversely, when $x_{p,r}=\frac{\ell_p}{n_r},\forall p,r$, i.e., $\mathrm x$ is a uniformly random assignment, its $\mathrm{L2Norm}$ becomes the smallest possible. Therefore, we can see that $\mathrm{L2Norm}$ encourages the assignments to be more randomized.
>
> **Regarding Figure 1.**
>
> In Figure 1, the ideal assignment (a) does not assign more papers to each reviewer than (b). First of all, for reviewers $r_4$ and $r_5$, the assignments in (a) and (b) are the same. For reviewers $r_1,r_2,r_3$, in (a), they have $\frac{1}{3}$ probability to review paper $p_1,p_2,p_3$ each. So each of them is assigned $1$ paper in total. In (b), this is also the case. For example, $r_1$ has $\frac{1}{2}$ probability to review $p_2$ and $p_3$ each. So $r_1$ is assigned $1$ paper in total. Therefore, we can see that each reviewer in (a) and (b) reviews the same number of papers.
>
> Again, we want to thank you for these insightful comments. We hope that the response above addresses most of your concerns about our paper and we would highly appreciate it if you can re-evaluate our paper based on this response.

---

> > ### Comment · Reviewer_625T · 2023-08-18
> > **Thanks for your response**
> >
> > I've read the response and other reviewers' comments carefully. The response addresses my concerns, particularly regarding the motivation of the proposed method. As a result, I decide to raise my score to 6.

---

### Official Review · Reviewer_euo8 · 2023-07-04

**Soundness:** 3 good
**Presentation:** 3 good
**Contribution:** 3 good
**Rating:** 6
**Confidence:** 2

**Summary:**

In this paper, the authors propose a new algorithm to improve randomness in peer-review paper assignment without sacrificing assignment quality. The authors first identify a set of good randomness properties in paper assignment besides the max probability used in the PLRA. The authors propose a simple and practical variation to PLRA by adding a concave perturbation function. Empirical evaluation on two real-world datasets demonstrates superior performance of the proposed method to PLRA.

**Strengths:**

1. The paper is well written with clear motivation and justification of the method. The identification of good randomness properties could motivate future research on random paper assignments.
2. The proposed algorithm is very practical and easy to implement as convex optimization which can be solved efficiently with commercial software.
3. The authors demonstrate the superiority of the proposed method both theoretically and empirically on two real-world peer-review datasets.


**Weaknesses:**

1. Though the authors provide comprehensive comparison to PLRA, it would be interesting to see comparison of the proposed method to other baselines, even the deterministic ones. Also, it would be interesting to provide “randomness” metrics from samples of assignments.
2. The authors motivate the requirement of randomness from 4 perspectives. It would be interesting to design more direct metrics or evaluation in terms of these 4 perspectives.
3. It would be better to include experiment and discussion on hyper-parameter sensitivity. As both PM-Q and PM-E contain the same number of parameters as in PLRA.


**Questions:**

N/A

**Limitations:**

Yes

---

> ### Author Rebuttal · Authors · 2023-08-06
>
> Thank you for spending time understanding our work and providing insightful comments. We are very glad to see your positive review and you recognize our work as superior in performance and very practical. Below we briefly respond to the weaknesses raised in your review.
>
> **Regarding comparison to other baselines.**
>
> We would like to highlight that the deterministic maximum matching algorithm can be seen as a special case of PLRA with $Q = 1$, and thus its performance is included in all the plots.
>
> Our experiments mainly evaluate the trade-offs between quality and randomness of PM and PLRA. Similar to the deterministic maximum matching algorithm, any deterministic algorithm will always produce deterministic assignments and thus will perform poorly in all randomness metrics.
>
> **Regarding direct randomness metrics to motivations.**
>
> In fact, two out of the four proposed metrics are direct choices for motivations in Section 1.
>
> **(i):** Average maximum probability ($\mathrm{AvgMaxp}$) is a direct choice for robustness to malicious behavior (Motivation 1). Depending on the model of malicious behavior, different metrics can fit into the motivation. The vanilla randomness metric $\mathrm{Maxprob}(\mathbf x)$ proposed in [22] corresponds to a model where the malicious reviewer-paper pair is a single worst-case pair chosen by Nature. However, if a particular paper $p$ is targeted, then in the worst case, the malicious reviewer targeting $p$ succeeds with probability $\max_{r}\lbrace x_{p,r}\rbrace$. Our new metric $\mathrm{AvgMaxp}(\mathbf x)$ calculates the average of these probabilities, corresponding to the model where a random paper $p$ is targeted with a worst-case malicious reviewer.
>
> **(ii):** Support size ($\mathrm{Support}$) is a direct choice for evaluation of alternative assignments (Motivation 2). The techniques for off-policy evaluation discussed in [21] can be utilized to estimate the review quality of another non-deployed assignment. Crucially, the precision of the estimation depends on the number of "positivity violations'' [21] between the deployed assignment $\mathbf x$ and the non-deployed assignment $\mathbf y$ of interest with positivity violation defined as an entry $(i,j)$ where $x_{i,j}=0$ but $y_{i,j}\ne 0$. Since a large number of alternative assignments $\mathbf y$ may be of interest, to minimize the number of positivity violations, we want to minimize the number of entries in $\mathbf x$ set to  $0$. This corresponds to maximizing $\mathrm{Support}(\mathbf x)$.
>
> That being said, we stress that sometimes, a direct randomness metric cannot be uniquely identified from a specific motivation alone. For example, as stated above, when we care about robustness to malicious behavior, the model of such behavior determines the appropriate  metric to use, and different models lead to different metrics. In practice, conference program chairs likely do not have a highly-specific model of malicious behavior in mind against which they wish to defend. As a result, we consider a “one-size-fits-all” approach in this paper, based on several general-purpose randomness metrics.
>
> **Regarding experiments on hyperparameters.**
>
> We do have a formal description of a principled way to choose hyperparameters in PM, including experiments about what happens when we choose different sets of hyperparameters. You can find it in Appendix A.2 (in the full paper included in the supplementary materials). We will add more description to the main paper in the next version.
>
> We again thank you for your understanding and recognition of our work, as well as for providing your insightful comments.

---

> > ### Comment · Reviewer_euo8 · 2023-08-13
> > **Thank you for the rebuttal**
> >
> > Thanks for the response. I have read the rebuttal and remain my initial evaluation.

---

### Official Review · Reviewer_iL2x · 2023-07-07

**Soundness:** 4 excellent
**Presentation:** 4 excellent
**Contribution:** 3 good
**Rating:** 8
**Confidence:** 3

**Summary:**

The paper presents an improvement on PLRA, an important (used in major reviewing deployments like AAAI and OpenReview) algorithm for randomizing reviewer assignments while preserving a prescribed fraction of assignment quality, by finding randomized assignments that increase additional randomization metrics of interest beyond bounding the probability a particular reviewer is assigned to a particular paper. These additional metrics help find solutions of comparable quality (as defined by PLRA) that address additional concerns such as increasing average randomness (intended to further guard against malicious behavior) and the support size (useful for better off-policy evaluation). The work is soundly evaluated theoretically and empirically.

**Strengths:**

The paper makes a significant contribution to reviewer assignment algorithms, an important topic relevant to the NeurIPS community. The proposed algorithm substantially improves on the previous algorithm with little to no apparent downside, and can be readily deployed. The work makes an original contribution and is presented with very high clarity.

**Weaknesses:**

The paper is very strong, but some minor weaknesses remain. The work relaxes the L2 norm constraint employed by PLRA to a soft constraint, but it is unclear how this relaxation might negatively harm results. The proposed algorithm appears novel in the context of reviewer assignment, but novelty in the context of related problems is less clear - it would be better to clarify the extent to which the technical contribution leverages known techniques (e.g., max flow) and how different this application is from related problems that have employed similar techniques in the past. The paper empirically tests on only 2 datasets, where it would seem easy to evaluate on additional datasets. See also my questions in the next section.

**Questions:**

- The proposed approach improves on PLRA but how does it compare to [20,29,30]?
- How much does relaxing the L2 norm to a soft constraint instead of the hard constraint used by PLRA negatively impact results?
- How were AAMAS and ICLR 2018 chosen, and why were additional datasets not also experimented with? Would the empirical results generalize?
- How can conference chairs prioritize certain proposed metrics over others? Is there a relative tradeoff?
- How/why was 95% picked for Table 1, versus also providing results for other percentages 80-100%?


**Limitations:**

The limitations could include more details on the limitations of the empirical experiments such as potential lack of generality from testing on only 2 datasets.

---

> ### Author Rebuttal · Authors · 2023-08-06
>
> Thank you for spending time understanding our work and providing insightful comments. We are very glad to see your positive review and that you recognize our work as significant, substantial, and ready to deploy. We respond to the questions and limitations raised in your review below.
>
> **Regarding the comparison to [20, 29, 30].**
>
> None of [20, 29, 30] considered randomized assignments. Indeed, [20, 29, 30] took different approaches to computing reviewer-paper assignments under different considerations. However, the three of them were restricted to deterministic assignments, i.e., assignments $\mathbf{x}\in\lbrace 0,1\rbrace ^{n_p\times n_r}$. Therefore, in terms of the proposed randomness metrics, these methods would perform very poorly. We then decide not to include these methods into our comparison.
>
> **Regarding the relaxation in L2 norm.**
>
> When we mention that the L2 norm relaxes the hard constraints used by PLRA, we intend to provide the intuition that a higher probability results in an increasingly larger loss in L2 norm. However, we do not use the L2 norm as a constraint in the optimization of PM. Instead, the L2 norm is used as a measure of randomness after PM produces a solution. We still keep the same hard constraints in PM as are used in PLRA. Therefore, the results are not negatively impacted by any relaxation in L2 norm.
>
> **Regarding the choice of datasets.**
>
> We used to run our experiments on various more datasets. In particular, we experimented with the following 6 datasets:
>
> - Preflib1 (00039-1 from [39]): 54 papers, 31 reviewers
> - Preflib2 (00039-2 from [39]): 52 papers, 24 reviewers
> - Preflib3 (00039-3 from [39]): 176 papers, 146 reviewers
> - AAMAS2015 (00037-1 from [39]): 613 papers, 201 reviewers
> - AAMAS2016 (00037-2 from [39]): 442 papers, 161 reviewers
> - ICLR2018 (from [26]): 911 papers, 2435 reviewers
>
> Among these 6 datasets, the first 5 are all bidding datasets, which means that for each paper-reviewer pair, there is a discrete level chosen from {"yes", "maybe", "no", "conflict"}. In contrast, ICLR2018 contains the text-similarity scores between the paper text and reviewers' past work. These scores are continuously-valued instead of discrete.
>
> Our experiments showed that on the 5 bidding datasets, the comparisons between PM and PLRA were always similar. Therefore, to optimize the presentation of our paper, we only kept the largest dataset (AAMAS2015) among these 5 datasets, together with ICLR2018. The empirical results did generalize to the other datasets we tested on and we are happy to add more experimental results for other datasets in the appendix in the next version.
>
> **Regarding trade-offs faced by conference chairs.**
>
> From Figures 3 and 4, we can see that essentially, the conference chairs are facing the trade-off between quality and randomness. If the conference chairs are willing to sacrifice more quality, PM will in return find an assignment with more randomness (with all randomness metrics improved simultaneously). To choose one point on this trade-off, in practice, the conference chairs could also plot it as in Figures 3 and 4, and choose a specific point on this curve based on their discretion.
>
> **Regarding Table 1.**
>
> Table 1 aims to show that the network-flow-based approximation takes less total CPU time with only minor degradation in performance. In terms of running time, the results for other percentages 80% - 100% will not be significantly different from 95%. Empirically, the degrees of degradation in performance are also similar within 80% - 100%. Therefore, we choose to only present the results of 95% since 5% is a reasonable sacrifice for robustness in real conferences. In this way, the readers will not be distracted by a large chunk of homogeneous experiment data.
>
> We again thank you for your understanding and recognition of our work, as well as for providing your insightful comments.

---

> > ### Comment · Reviewer_iL2x · 2023-08-16
> > **Rebuttal response**
> >
> > I thank the authors for addressing my questions. I am somewhat concerned about the selective reporting; can how "similar" the withheld results are be quantified? While I understand the concerns about presentation, it is important to report full results. It is common for papers to have tables with results across many datasets, and this would strengthen the current work.

---

> > > ### Author Response · Authors · 2023-08-17
> > > **Experiment Results on Datasets Mentioned in Rebuttal**
> > >
> > > Thank you for reading our rebuttal and giving this response.
> > >
> > > To address your concern, we report in this response additional experiment results. As we are not allowed to upload pictures at this stage, we will first provide you with our experiment results on Preflib1 summarized in the following tables.
> > >
> > > **Maxprob (↓) vs. Relative Quality on Preflib1**
> > >
> > > | Relative Quality | 80%    | 85%    | 90%    | 95%    | 98%    | 99%    | 99.5%  | 100%   |
> > > | ---------------- | ------ | ------ | ------ | ------ | ------ | ------ | ------ | ------ |
> > > |PLRA|0.4482|0.5410|0.6611|0.8184|0.9277|0.9639|0.9824|1.0000|
> > > |PM-Q|0.4482|0.5410|0.6611|0.8184|0.9277|0.9639|0.9824|1.0000|
> > > |PM-E|0.4482|0.5410|0.6611|0.8184|0.9277|0.9639|0.9824|1.0000|
> > >
> > > **AvgMax (↓) vs. Relative Quality on on Preflib1**
> > >
> > > | Relative Quality | 80%    | 85%    | 90%    | 95%    | 98%    | 99%    | 99.5%  | 100%   |
> > > | ---------------- | ------ | ------ | ------ | ------ | ------ | ------ | ------ | ------ |
> > > |PLRA|0.4482|0.5410|0.6611|0.8184|0.9277|0.9639|0.9824|1.0000|
> > > |PM-Q|0.4196|0.4942|0.5865|0.7013|0.7792|0.8055|0.8186|0.8321|
> > > |PM-E|0.4201|0.4946|0.5868|0.7018|0.7799|0.8062|0.8194|0.8328|
> > >
> > > **Support (↑) vs. Relative Quality on Preflib1**
> > >
> > > | Relative Quality | 80%    | 85%    | 90%    | 95%    | 98%    | 99%    | 99.5%  | 100%   |
> > > | ---------------- | ------ | ------ | ------ | ------ | ------ | ------ | ------ | ------ |
> > > |PLRA|386|331|276|224|216|218|218|162|
> > > |PM-Q|1178|1060|868|761|740|723|727|651|
> > > |PM-E|1188|1058|870|761|737|730|727|640|
> > >
> > > **Entropy (↑) vs. Relative Quality on Preflib1**
> > >
> > > | Relative Quality | 80%    | 85%    | 90%    | 95%    | 98%    | 99%    | 99.5%  | 100%   |
> > > | ---------------- | ------ | ------ | ------ | ------ | ------ | ------ | ------ | ------ |
> > > |PLRA|137.81|110.38|79.91|47.24|30.01|21.26|12.92|0.00|
> > > |PM-Q|252.57|221.67|190.38|161.57|143.42|135.43|131.21|125.49|
> > > |PM-E|252.86|221.82|190.49|161.65|143.56|135.56|131.40|125.37|
> > >
> > > **L2 Norm (↓) vs. Relative Quality on Preflib1**
> > >
> > > | Relative Quality | 80%    | 85%    | 90%    | 95%    | 98%    | 99%    | 99.5%  | 100%   |
> > > | ---------------- | ------ | ------ | ------ | ------ | ------ | ------ | ------ | ------ |
> > > |PLRA|8.36|9.13|10.05|11.11|11.88|12.22|12.47|12.73|
> > > |PM-Q|6.75|7.47|8.23|9.06|9.64|9.87|9.99|10.12|
> > > |PM-E|6.75|7.47|8.23|9.06|9.65|9.87|9.99|10.12|
> > >
> > > The above results are similar to Figures 2(a) and 3, in particular, both versions of PM achieve exactly the same performance with PLRA on Maxprob while improving significantly on the other randomness metrics.
> > >
> > > As the response is limited to 5000 characters, we are not able to provide the results on the other datasets (Preflib2, Preflib3 and AAMAS2016) in this response. But if you are interested, we are happy to give another 3 responses regarding these datasets.
> > >
> > > We agree with your point that having the results on more datasets written in the paper would strengthen the current work, and we will include these results in the next version.

---

### Official Review · Reviewer_FQ98 · 2023-07-10

**Soundness:** 3 good
**Presentation:** 2 fair
**Contribution:** 2 fair
**Rating:** 5
**Confidence:** 4

**Summary:**

This work is concerned with randomness in paper matching for peer review. It provides normative motivations for the value of randomness in this process and proposes a family of generalizations of the Probability Limited Randomized Assignment (PLRA) paper matching framework of Jecmen et al. This generalization, which the authors term Perturbed Maximization, entails replacing the linear objective in the PLRA linear programming problem with a concave objective which increasingly discounts paper-reviewer matchings with higher probabilities. The authors prove comparisons between the two methods under two distributions of paper-reviewer similarity scores. Experimentally, the authors consider two choices of discounting (perturbation) functions, demonstrate that solving this convex problem is tractable in practice, and present the effect of deploying their method for two past conferences, according to multiple measures of randomness.

**Strengths:**

This work proposes and studies a natural generalization of the PLRA paper matching framework. The generalization and analysis are particularly suited to practical use, in the generalization can be arbitrarily incremental, and the PLRA framework has been used in practice and serves as a benchmark for the experiments.

From a theoretical perspective, it is interesting to identify combinatorial optimization problems/applications for which randomness is a desirable property, and to study how randomness this trades off against optimality.

The experimental results seem to demonstrate that, at least for some paper matching instances, the framework could be used to generate matchings with comparable quality in a much more randomized way.

**Weaknesses:**

Theoretical results: I have a difficult time understanding the settings and claims of the theoretical analysis provided, and I am suspicious of the implications of the analysis to understanding PM as compared to PLRA. The quantity c in Theorem 3 is unspecified, and the claims seem to suggest that the models permit either the same or somehow degenerate solution sets for the two algorithms. In any case, more explanation of these claims seems warranted.

Randomness metrics: the motivation for the proposed randomness metrics for the distribution x in Section 3 is very thin. Some references for these being useful measures for a fractional 'matching' in this application or elsewhere would be welcome. But there seems to be a deeper mismatch between these measures and the reasons randomness is desirable outlined in Section 1. Properly, (and as observed in the discussion of entropy), x is not a distribution; instead the fractional assignment x together with a dependent rounding scheme together induce a distribution over reviewer-paper assignments.  The randomness of this distribution seems arguably more relevant to most of the randomness motivations provided in Section 1; for instance, depending on the dependent rounding scheme, a fractional assignment x with Q=0.1 could correspond to sampling one of only 10 final assignments, allowing easy reviewer de-anonymization. I think that calculating (or estimating) these randomness metrics for the assignment distribution would lead to a superior analysis.

**Questions:**

Randomness metrics:
	Can you remark on which metrics in Section 3 would be natural choices for any of the randomness motivations in Section 1, and whether you agree that in any cases the corresponding metrics for the induced assignment distribution would be more relevant?
	Alternatively, do you think there is good motivation for restricting your randomness metrics to ones that are linear over the fractional assignments of individual papers (as all except L2 are)?
	Do the authors of [22] mention or consider any metrics other than Maxprob?

Theoretical results:
	What is the scalar c in the statement of Theorem 3? Is it an unspecified constant, or does the statement hold for every c?
	If part (a) of Theorems 2 and 3 holds for all pairs of points in the solution set, the claimed Quality inequality seems to suggest that the solution set of PM is contained in the solution set for PLRA. This in turn raises two questions: first, given that the PM objective is different, wouldn't this suggest that these theoretical settings are in a sense too degenerate (have too many equally optimal solutions/too few nearby suboptimal ones) to distinguish between the two methods? Second, do the remaining claimed inequalities suggest that PM identifies a unique point in PLRA(Q) which simultaneously minimizes/maximizes those randomness measures within the set PLRA(Q)?

On line 333, how does Figure 4 show that PM sacrifices Maxprob relative to PLRA?

**Limitations:**

The choice of perturbation function seem ad hoc, and it is not obvious why the authors decided to apply the perturbation function to the full range [0,1] instead of to the range [0,Q]. For the perturbation functions studied here this does not matter, but in general it could. There are three places in the narrative where the authors present a range of options: (i) motivations for randomness in paper matching, (ii) measures of randomness, and (ii) choices of perturbation function. It would have been interesting to see some discussion of how choices of (i) might motivate different choices of (ii), particularly with regard to motivations 2 and 4, and some discussion of how different choices of dependent rounding scheme given the fractional assignment affect these objectives. But it would have been particularly nice to see some motivation of how choices of (ii) inform choices of (iii). For instance, are there informal or heuristic arguments that would suggest that one form of perturbation function would be particularly suited to maximizing the entropy objective?

The theoretical analysis is a little unsettling. Either the theoretical similarity models considered seem inapt to the purpose of comparing the two methods, or the theoretical claims could use significantly more contextualization.

Certain aspects of the experimental section could be clearer. In particular, it would be helpful to include a slightly more explicit description of the hyperparameters that are being fit, and the order in which they are being fixed. In my opinion this omission makes the presentation and interpretation of Figure 2 particularly misleading, since the form it takes and the insights offered are directly determined by the value assigned to the hyperparameter delta, which is not mentioned.

---

> ### Author Rebuttal · Authors · 2023-08-03
>
> Thank you for spending time understanding our work and providing insightful comments. We are happy you find our contributions relevant, interesting, and practical. Below we respond to the questions and limitations raised in your review, hoping to address your concerns about our paper.
>
> **Regarding randomness metrics.**
>
> In Section 3, two out of the four metrics are natural choices for motivations in Section 1.
>
> **(i):** Average maximum probability ($\mathrm{AvgMaxp}$) is a natural choice for robustness to malicious behavior (Motivation 1). Depending on the model of such behavior, different metrics can fit into the motivation. The vanilla randomness metric $\mathrm{Maxprob}(\mathbf x)$ proposed in [22] corresponds to a model where the malicious reviewer-paper pair is a single worst-case pair chosen by Nature. However, if a particular paper $p$ is targeted, then in the worst case, the malicious reviewer targeting $p$ succeeds with probability $\max_{r}\lbrace x_{p,r}\rbrace$. Our new metric $\mathrm{AvgMaxp}(\mathbf x)$ calculates the average of these probabilities, corresponding to the model where a random paper $p$ is targeted with a worst-case malicious reviewer.
>
> **(ii):** Support size ($\mathrm{Support}$) is a natural choice for evaluation of alternative assignments (Motivation 2). The techniques for off-policy evaluation [21] can be utilized to estimate the review quality of another non-deployed assignment. Crucially, the precision of the estimation depends on the number of "positivity violations'' [21] between the deployed assignment $\mathbf x$ and the non-deployed assignment $\mathbf y$ of interest, with positivity violation defined as an entry $(i,j)$ where $x_{i,j}=0$ but $y_{i,j}\ne 0$. Since a large number of alternative assignments $\mathbf y$ may be of interest, to minimize the number of positivity violations, we want to minimize the number of entries in $\mathbf x$ set to $0$. This corresponds to maximizing $\mathrm{Support}(\mathbf x)$.
>
> While alternative metrics that operate on the full assignment distribution (as opposed to simply the fractional assignment) are possible, such metrics are only beneficial if the joint distribution of reviewer-paper assignments is of special interest. In the above two cases, the corresponding metrics for the induced assignment distribution are equivalent to the metrics for the fractional assignment that we use, and thus they are equally relevant. As such, we focus in this work on metrics that operate only on these marginal probabilities.
>
> We are not deliberately restricting our randomness metrics linear over fractional assignments of individual papers. We choose the first two metrics because of the reasons above. The remaining two metrics (Entropy and L2 Norm) are chosen based on general principles: (i) Entropy characterizes the uncertainty of a distribution and (ii) L2 Norm places a larger penalty on entries with higher probability. These two metrics can be viewed as additional ways of characterizing the distance from a uniform random assignment, which is the most random assignment.
>
> [22] did not mention or consider any metrics other than $\mathrm{Maxprob}$.
>
> **Regarding theoretical results.**
>
> Theorem 3 holds for every $c>0$.
>
> For PLRA, there are indeed many equally optimal solutions in the theoretical settings. This fact does not imply that the settings are too degenerate since PLRA's optimization does allow many equally optimal solutions in many cases. Recall that in PLRA, the objective function is $\sum_{p,r}x_{p,r}S_{p,r}$. When $S_{p_1,r_1}=S_{p_2,r_2}$, adjusting the solution by adding $c$ to $x_{p_1,r_1}$ and subtracting $c$ from $x_{p_2,r_2}$ does not affect the objective function. Therefore, when many entries in the matrix $\mathbf S$ share the same value, one can imagine that there will be many ways to adjust an optimal solution into another optimal solution while preserving feasibility. In practice, the phenomenon of many entries in the matrix $\mathbf S$ sharing the same value is possible (e.g., when the matrix $\mathbf S$ is constructed from discrete reviewer bids and subject areas only). The answer to your second question is "yes". Among these many optimal solutions of PLRA, PM identifies a subset of solutions that simultaneously minimize/maximize those randomness measures within the set.
>
> **Regarding Line 333.**
>
> Figure 4 does not show PM sacrifices $\mathrm{Maxprob}$ relative to PLRA. Figure 2b (which is also mentioned in Line 333) shows it instead. In Figure 2b, you can see that the yellow and green lines (showing PM’s $\mathrm{Maxprob}$) are slightly higher than the blue line (showing PLRA’s $\mathrm{Maxprob}$).
>
> **Regarding the choice of perturbation functions.**
>
> For (i) motivations for randomness (ii) measures of randomness and (iii) choices of perturbation function, we clarify how (i) might motivate (ii) above. For how choices of (ii) inform choices of (iii), we do have a formal discussion in Appendix C.1 (in supplementary materials). For instance, we specify one form of perturbation function which is exactly suited to maximizing the entropy objective there (Theorem 4).
>
> In the main paper, we choose two simpler perturbation functions to clearly convey the intuition behind PLRA, while keeping things easy to understand.
>
> **Regarding the experiments.**
>
> We do have a formal description of a principled way to choose hyperparameters in PM, including how we pin down $\delta$ in Appendix A.2 (in supplementary materials). In short, as we increase $\delta$, we allow PM-Q and PM-E to find more random solutions at a cost of larger $\mathrm{Maxprob}$. To balance the gain and loss, we choose $\delta$ that maximizes a linear combination of them. We will add more description to the main paper in the next version.
>
> Again, we want to thank you for these insightful comments. We hope that the response above addresses most of your concerns and we would highly appreciate it if you can re-evaluate our paper based on this response.

---

### Author Rebuttal · Authors · 2023-08-06

We firstly want to thank all the reviewers for spending the time to understand our work and provide insightful comments.

We are happy to see that all the reviewers concurred that our proposed method “has better performance” (reviewer 625T) in randomness with “little to no apparent downside (iL2x)" and is "particularly suited to practical use" (FQ98). Also, most of the reviewers concurred that our paper is "well written" (euo8) and ”easy to understand” (625T).

We are also glad that reviewers recognized that our “theoretical analyses regarding how the proposed PM algorithm outperforms PLRA are sufficient" (625T) and our work tackles "important topic relevant to the NeurIPS community" (iL2x).

Below we provide detailed responses to each reviewer’s questions.

---

### Decision · Program_Chairs · 2023-09-21

**Decision:**

Accept (spotlight)

**Comment:**

All of the reviewers agree that the paper makes significant contributions to the well-motivated problem of assigning peer-reviewers to manuscripts. The paper builds on work by Jecmen et al, and introduces discounting functions that can controllably increase randomness in paper assignments. During the discussion phase, the authors reported additional experiment results which should be included in an appendix. The authors also clarified several points about the theory, design choices (randomness metrics, perturbation functions, experiment datasets) and Figure 1 which would substantially strengthen the paper's exposition if included in the revised manuscript.